# Membrane-assisted assembly and selective secretory autophagy of enteroviruses

Selma Dahmane[1,2,3], Adeline Kerviel[4], Dustin R. Morado ●[5], Kasturika Shankar[1,2,3], Björn Ahlman[1,2,3], Michael Lazarou ●[6], Nihal Altan-Bonnet ●[4] & Lars-Anders Carlson ●[1,2,3] ✉

Enteroviruses are non-enveloped positive-sense RNA viruses that cause diverse diseases in humans. Their rapid multiplication depends on remodeling of cytoplasmic membranes for viral genome replication. It is unknown how virions assemble around these newly synthesized genomes and how they are then loaded into autophagic membranes for release through secretory autophagy. Here, we use cryo-electron tomography of infected cells to show that poliovirus assembles directly on replication membranes. Pharmacological untethering of capsids from membranes abrogates RNA encapsidation. Our data directly visualize a membrane-bound half-capsid as a prominent virion assembly intermediate. Assembly progression past this intermediate depends on the class III phosphatidylinositol 3-kinase VPS34, a key host-cell autophagy factor. On the other hand, the canonical autophagy initiator ULK1 is shown to restrict virion production since its inhibition leads to increased accumulation of virions in vast intracellular arrays, followed by an increased vesicular release at later time points. Finally, we identify multiple layers of selectivity in virus-induced autophagy, with a strong selection for RNA-loaded virions over empty capsids and the segregation of virions from other types of autophagosome contents. These findings provide an integrated structural framework for multiple stages of the poliovirus life cycle.

Enteroviruses are a major genus of positive-sense RNA viruses within the Picornaviridae family. They cause a wide variety of human diseases such as poliomyelitis (poliovirus), related acute flaccid myelitis conditions (e.g., EV-D68), and viral myocarditis (Coxsackievirus B3). The enterovirus particle is a non-enveloped particle of ~30 nm diameter, encapsidating a single-stranded RNA genome of about 7500 nucleotides.

Upon infection, enteroviruses rapidly remodel cytoplasmic membranes to create an optimal environment for virus replication[1–3]. This starts with disassembly of the secretory pathway, in particular the Golgi apparatus as seen by dispersion of Golgi marker proteins throughout the cytoplasm[4]. Ensuing single-membrane tubules and vesicles are the site of viral RNA replication (along with later double-membrane vesicles), but little is known about the site of virion assembly[3,5]. The viral genome has preferential interaction sites with the capsid but contains no known high-affinity packaging signal sufficient for RNA loading into capsids[6,7]. This suggests that virion assembly may take place in immediate vicinity of RNA production sites to ensure specific RNA encapsidation, which is supported by the finding that the viral membrane-bound helicase 2C interacts with the capsid protein VP3[7,8].

[1]Department of Medical Biochemistry and Biophysics, Umeå University, Umeå, Sweden. [2]Wallenberg Centre for Molecular Medicine, Umeå University, Umeå, Sweden. [3]The Laboratory for Molecular Infection Medicine Sweden (MIMS), Umeå University, Umeå, Sweden. [4]Laboratory of Host-Pathogen Dynamics, National Heart Lung and Blood Institute, National Institutes of Health, Bethesda, MD, USA. [5]Department of Biochemistry and Biophysics, Science for Life Laboratory, Stockholm University, Stockholm, Sweden. [6]Department of Biochemistry and Molecular Biology, Biomedicine Discovery Institute, Monash University, Melbourne, Australia. ✉e-mail: lars-anders.carlson@umu.se

A biochemically distinct late stage of replication starts around 6 h post-infection (h p.i.). It is characterized by the lipidation of host proteins belonging to the LC3 subfamily of ATG8s, a hallmark of the autophagy pathway[9]. Induction of LC3 lipidation in infected cells is mediated by the viral protein 2BC and is independent of the ULK1/ULK2 protein kinases that initiate canonical autophagy[9,10]. At this late stage, the single-membrane tubes and vesicles are observed to be replaced by (suggested to transform into) double-membrane structures reminiscent of autophagic membranes[11,12]. The inherently non-enveloped picornaviruses have recently been shown to leave cells non-lytically as groups of virions contained in LC3-positive lipid vesicles, utilizing the secretory autophagy pathway[13–17]. It thus seems plausible that autophagy-like double-membrane structures observed in infected cells relate to non-lytic virus egress. However, conventional EM sample preparation is too destructive to macromolecular structure to allow an in situ analysis of autophagosome contents—the experiment needed to directly test this hypothesis.

To shed light on how enteroviruses are assembled and packaged into autophagosomes, we took advantage of recent advances in focused-ion-beam milling and cryo-electron tomography[18–22]. The in situ structures of poliovirus-infected cells revealed that enteroviruses assemble directly on replication membranes. Completion of virus assembly requires the host lipid kinase VPS34, and RNA loading into virions correlates with their membrane tethering. Inhibiting the initiation of canonical autophagy surprisingly increased virion production and release. The cryo-electron tomograms further revealed that virus-induced autophagy has a striking degree of selectivity, selecting RNA-containing virions over empty capsids, and segregating virions from other types of autophagosome contents.

## Results

### Cryo-electron tomography reveals that poliovirus RNA loading correlates with capsid tethering to membranes

To investigate enterovirus assembly in situ, we infected HeLa cells with poliovirus type 1 and imaged the cytoplasm with cryo-electron tomography at different time points post infection (Supplementary Fig. 1). At 3 h p.i. first new virions had already assembled in the cells (Supplementary Fig. 2a–d). Tomograms recorded at 6 h p.i. showed a more starkly remodeled cytoplasm (Fig. 1a, b, Supplementary Movie 1). Compared to uninfected cells and 3 h p.i., there was a significant increase in both open cup-shaped structures resembling phagophores and closed double-membrane vesicles (Fig. 1a, b, Supplementary Fig. 2a–g). These membranes will collectively be referred to as autophagy-like membranes (ALMs).

From the tomograms we could clearly distinguish empty capsids from RNA-loaded virions (Fig. 1a, b). At 6 h p.i. the cytoplasmic concentration of empty and RNA-loaded particles was on average 7 and 20 times higher than at 3 h p.i., respectively, as measured by a template matching procedure (Fig. 1e). Strikingly, both empty capsids and RNA-loaded virions were frequently tethered to membranes through macromolecular complexes (Fig. 1a–d, yellow arrowheads). The tether appeared to have a defined size and keep the virions at a defined distance from the membrane. A subtomogram average of 179 tethered virions revealed the tether as having a height of ~6 nm, width ~12 nm and an approximate molecular mass of ~230 kDa (Fig. 1f, Supplementary Fig. 2j). At 6 h p.i., virions were tethered to both single-membrane tubes and vesicles (SMs) and ALMs, and the fraction of empty vs. RNA-loaded virions was similar on both types of membranes (Fig. 1f). We noticed that virions were only tethered to the outer face of ALMs and SMs, whereas virions engulfed by ALMs had lost the tether (Fig. 1a, red arrow and Fig. 1g).

The visualization of capsids tethered to replication membranes suggested that capsid RNA loading took place on membranes rather than in the cytosol. To test this we used (5-3,4-dichlorophenyl)-methylhydantoin (henceforth Hydantoin), an antiviral drug that inhibits RNA loading of PV capsids[23]. Cells were infected with PV and treated with Hydantoin at a concentration of 50 μg/ml that did not interfere with viral RNA replication[23] (Supplementary Fig. 4h) before processing for cryo-ET at 6 h p.i. This revealed both an increase in the fraction of empty capsids to ~70% and a ~3-fold decrease in the fraction of tethered capsids in Hydantoin-treated cells (Fig. 1h–k). Notably, that the abundance of SMs and ALMs remained unchanged (Supplementary Fig. 2h). Together, these data show that newly assembled poliovirus capsids are tethered to the cytoplasmic face of SMs and ALMs and tethering facilitates viral RNA encapsidation.

### Enterovirus capsid assembly takes place on membranes and requires VPS34 activity

Tomograms of infected cells at 6 h p.i. frequently contained novel structures that had a size and shape consistent with partial capsids (Fig. 2a–e, Supplementary Fig. 3a, b). Strikingly, in 15 tomograms of infected cells, 96% of these bona fide capsid intermediates were membrane-associated (58% SM, 38% ALM) whereas only 4% were found in the cytoplasm (Fig. 2f). The capsid intermediates were clearly thinner than, and often discontinuous with, the replication membranes, ruling out that this was an extension of the lipid membrane (Fig. 2b–d, Supplementary Fig. 3a–e). The localization of capsid assembly intermediates to membranes was supported by an immunofluorescence co-staining using an anti-LC3 antibody and the monoclonal antibody A12 which specifically recognizes the canyon region of assembled capsids and capsomers[24], showing a high degree of colocalisation (Supplementary Fig. 3f, g). The capsid intermediates contained variable luminal densities and were observed at different angles to the membranes (Fig. 2b–e). Due to this structural variability, we characterized them by a simple angle of closure, which resulted in a unimodal distribution with an average of 169°, i.e., closely corresponding to half a capsid (Fig. 2g). The clear clustering around a single value indicates that the membrane-bound capsid intermediate is a single, or a set of closely related, molecular species.

Given the frequent association of partial and complete capsids with ALMs, we inhibited autophagy using two selective inhibitors: MRT68921 and Vps34-IN1. MRT68921 inhibits the ULK1/ULK2 protein kinases that initiate canonical autophagy[25], whereas Vps34-IN1 inhibits the lipid kinase VPS34 that produces phosphatidylinositol-3-phosphate (PI(3)P) on the growing phagophore membrane[26]. Consistent with previous studies, PV-induced ULK-independent LC3 lipidation at 6 h p.i.[10]. However, co-treating infected cells with both inhibitors decreased LC3 lipidation (Supplementary Fig. 4a). Tomograms of co-treated cells revealed reduced membrane proliferation and a two-orders-of-magnitude decrease in both empty and RNA-loaded capsids (Fig. 3a–e; Supplementary Fig. 4b and Supplementary Movie 2). However, the number and distribution of membrane-bound capsid intermediates were unaffected (Fig. 3e; Supplementary Fig. 4c, d). The stalled progress from capsid intermediates to full capsids meant that we could observe a rare, more advanced assembly stage that may represent the transition from the half-capsid intermediate to a complete membrane-tethered virion (Fig. 3d). Moreover Vps34-IN1 treatment alone had a similar effect as the combination of inhibitors (Supplementary Fig. 4e–h). The decrease in intracellular assembled virus was mirrored by a decrease in virus release from VPS34-inhibited cells (Fig. 3f, Supplementary Fig. 4k). Notably, VPS34 inhibition caused a decrease in intracellular viral RNA, suggesting that a reduction in viral RNA replication might be the inhibitory mechanism of Vps34-IN1 on capsid assembly (Supplementary Fig. 4h).

Since VPS34 inhibition decreased LC3 lipidation, we sought to determine if Vps34-IN1's effect on virus assembly was mediated by ATG8 family proteins. We infected CRISPR-generated triple knock out (3KO) cells of the LC3 subfamily, and 3KO cells of the GABARAP subfamily and imaged them at 6 h p.i. The only significant change to

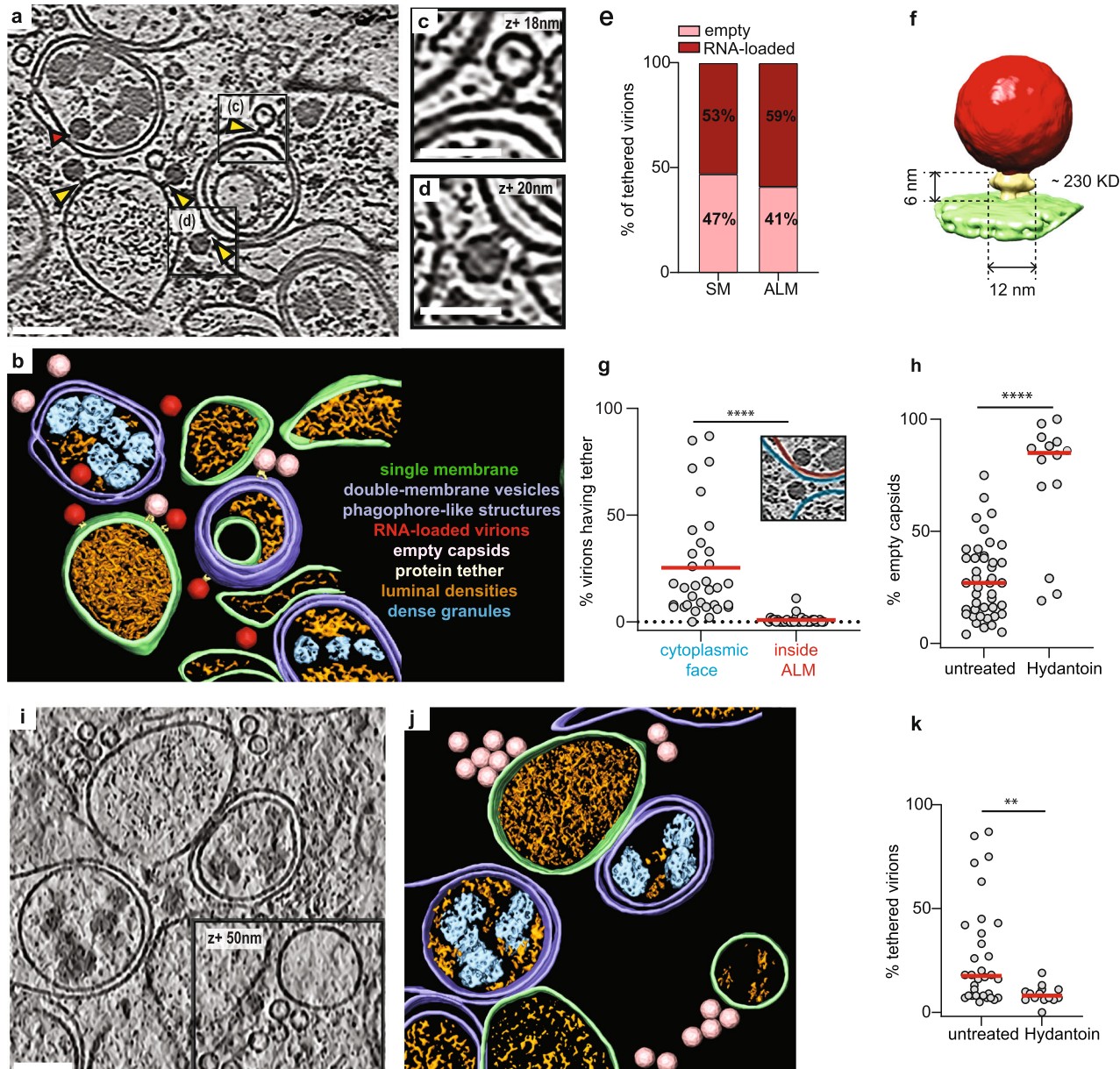

**Fig. 1 | Cryo-electron tomography allows the visualization of poliovirus replication and assembly sites in situ. a** Slice through a representative cryo-electron tomogram of a lamella milled through a PV-infected cell at 6 h p.i., revealing PV-induced SM (single membranes) and ALM (autophagy-like membranes) proliferation. Yellow arrowheads indicate densities tethering intracellular empty capsids and darker RNA-loaded virions to membranes. Red arrowhead indicates a virion enclosed inside a DMV (double-membrane vesicle), proximal but not tethered to the membrane. **b** Segmentation of the tomogram presented in (**a**). Color labels are defined for each structure. Empty capsids (pink) and RNA-loaded virions (red) are represented by their subtomogram averages. **c**, **d** Magnified view of an empty capsid and RNA-loaded virion tethered to a DMV shown in (**a**) with black boxes. **e** Percentage of empty (*n* = 223) and RNA-loaded virions (*n* = 281) observed on SM and ALM. **f** Subtomogram average of the tethered viral capsid at 70 Å resolution. Its height and width are marked, and its molecular mass is estimated to be ~230 kDa.

**g** Percentage of (*n*) virions observed on the outside (*n* = 520, blue) and inside (*n* = 22, red) of ALM having a visible tether to a membrane as indicated in the inset. **h** Percentage of empty capsids observed in (*n*) tomograms of untreated (*n* = 43) and Hydantoin-treated cells (*n* = 14) at 6 h p.i., as measured by template matching. Horizontal lines represent the average. **i** Cryo-electron tomogram of a PV-infected, Hydantoin-treated cell at 6 h p.i., containing several empty capsids which are not tethered to the surrounding membranes. **j** Segmentation of the tomogram in (**i**). Color labels for each structure are the same in (**b**) and empty capsids are represented by their subtomogram average. **k** Percentage of tethered virions as observed in (*n*) cryo-tomograms of untreated (*n* = 33) and Hydantoin-treated (*n* = 14) cells at 6 h p.i. Horizontal lines represent the average. In all graphs, each dot corresponds to one tomogram analyzed (see also Supplementary Table 2). Statistical significance by unpaired two-tailed Student's *t* test: **g** ****p < 0.0001, **h** **p = 0.0138, **k** **p = 0.0089. Scale bars: **a–i** 100 nm, **c, d** 50 nm.

---

membrane structures was a decrease in ALMs in infected LC3 3KO cells (Supplementary Fig 4i). This was paralleled by a decrease of intracellular virions in LC3 KO cells (Fig. 3g). Interestingly, those areas of LC3 3KO cytoplasm that still contained ALMs also contained virions, whereas areas with large SMs were devoid of virions (Fig. 3h, i). By comparison, GABARAP 3KO cells still robustly accumulated ALMs

(Fig. 3j), and there was less change in intracellular virus concentration upon GABARAP deletion (Fig. 3g). Together this indicates that VPS34 activity, rather than any individual ATG8 protein, are necessary for enterovirus assembly. This notion was strengthened by the observation that VPS34 inhibition further reduced virus release from LC3 3KO cells, similar to its effect on wild-type cells (Fig. 3k).

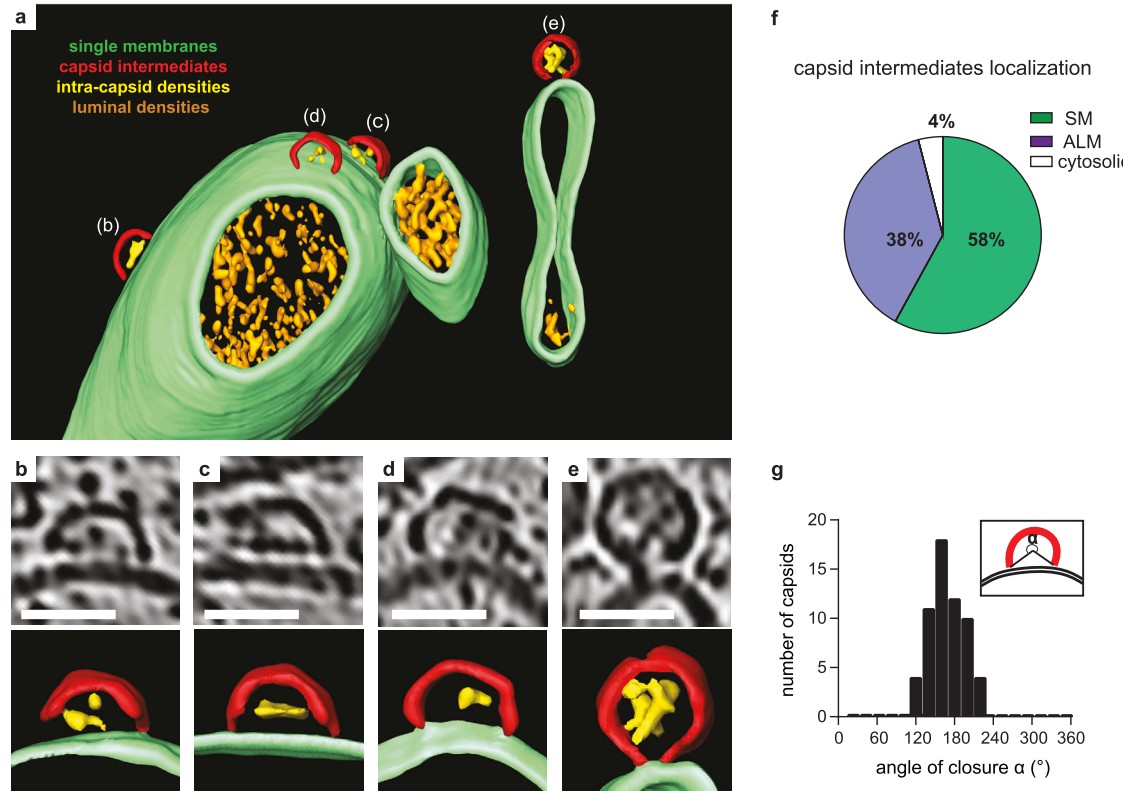

**Fig. 2 | A membrane-bound capsid intermediate. a** 3D segmentation of a tomogram showing capsid assembly intermediates, containing luminal densities, directly bound to single-membrane tubules (corresponding slice in Supplementary Fig. 3a, b). **b–e** Zoomed tomogram slices and segmentations of the capsids assembly intermediates marked in (**a**). **f** Percentage of capsid assembly intermediates found on SMs, ALMs or not associated with membranes, as counted in 29 tomograms at 6 h p.i. **g** Distribution of capsid intermediate closures (α), as defined in the inset, measured at 6 h p.i. Average closure was 169° (SD = 26°, *n* = 59). Scale bars: 50 nm.

In summary, enterovirus capsid assembly takes place on membranes with a prominent half-capsid intermediate, and activity of the lipid kinase VPS34 is required for assembly to progress beyond this intermediate.

## Inhibition of ULK1 leads to formation of intracellular virus arrays and increased vesicular release

Although inhibition of the canonical autophagy initiators ULK1 and ULK2 did not affect PV-induced LC3 lipidation or the generation of ALMs (Fig. 4a, b, Supplementary Fig. 5a), we found that ULK-inhibited cells contained large cytoplasmic arrays of virions (Fig. 4a, b, Supplementary Movie 3). The arrays contained several hundred virions, virtually all RNA-loaded, in what seemed to be a close-packing arrangement. Virus arrays were also visible in freeze-substituted sections, which allowed sampling of a larger number of cells (Supplementary Fig. 5c–e). Arrays were found in 3% of untreated cells and 56% of ULK-inhibited cells (Fig. 4c), a significantly higher fraction, showing that ULK inhibition indeed upregulates intracellular virus array formation. Reflecting the fact that virions in arrays are not membrane-proximal, the fraction of membrane-tethered virions was reduced in ULK-inhibited cells (Supplementary Fig. 5b).

The formation of virus arrays may either be due to a defect in virion release from ULK-inhibited cells, or the arrays may be part of an increased intracellular virion pool en route to release. The 'release scenario' was supported by the observation that RNA-loaded virions were abundant in ALMs in ULK-inhibited cells (Fig. 4d–g). Multilamellar structures, previously reported to be occasionally present in PV-infected cells[12], were frequently seen in ULK-inhibited cells. To further discriminate between these two scenarios, we measured released infectious virus from ULK-

inhibited cells at different time points and compared it to untreated cells. While extracellular PV titers were unchanged at 6 h p.i., we measured one order of magnitude increase over untreated cells at 8 h p.i. (Fig. 4h). To determine if the increased virus release at late time point still took place through secretory autophagy, we isolated the extracellular vesicular fraction. A strong increase in capsid protein as well as lipidated LC3 from ULK-inhibited cells confirmed that the virions were released in vesicles positive for LC3 (Fig. 4i). Imaging the periphery of non-FIB-milled, ULK-inhibited, and PV-infected cells we were also able to record a cryo-electron tomogram of extracellular vesicles containing RNA-loaded virions just outside the PM (Fig. 4j–n). Taken together, ULK activity in infected cells is not necessary for downstream autophagy processes such as LC3 lipidation but instead appears to put a break on the intracellular accumulation and vesicular release of virions.

## Selective packaging of RNA-loaded virion and contents segregation in autophagy-like membranes

The observation that virions are tethered to the outside of ALMs but not to the inside (Fig. 1g) indicated that cryo-ET can provide structural insights into the process of virion packaging into autophagy-like membranes. Indeed, the tomograms of PV-infected cells at 6 h p.i. showed several stages of autophagic engulfment of virions, ranging from wide open phagophores to completely sealed DMVs (Fig. 5a–e). We then re-evaluated the statistics of empty capsids and RNA-loaded virions at 6 h p.i. taking particle location (outside or inside ALMs) into consideration. Strikingly, this revealed that autophagic engulfment strongly selects for RNA-loaded virions. Outside ALMs, empty capsids represent 23% of particles, whereas only 1% of particles in ALMs were empty capsids (Fig. 5f).

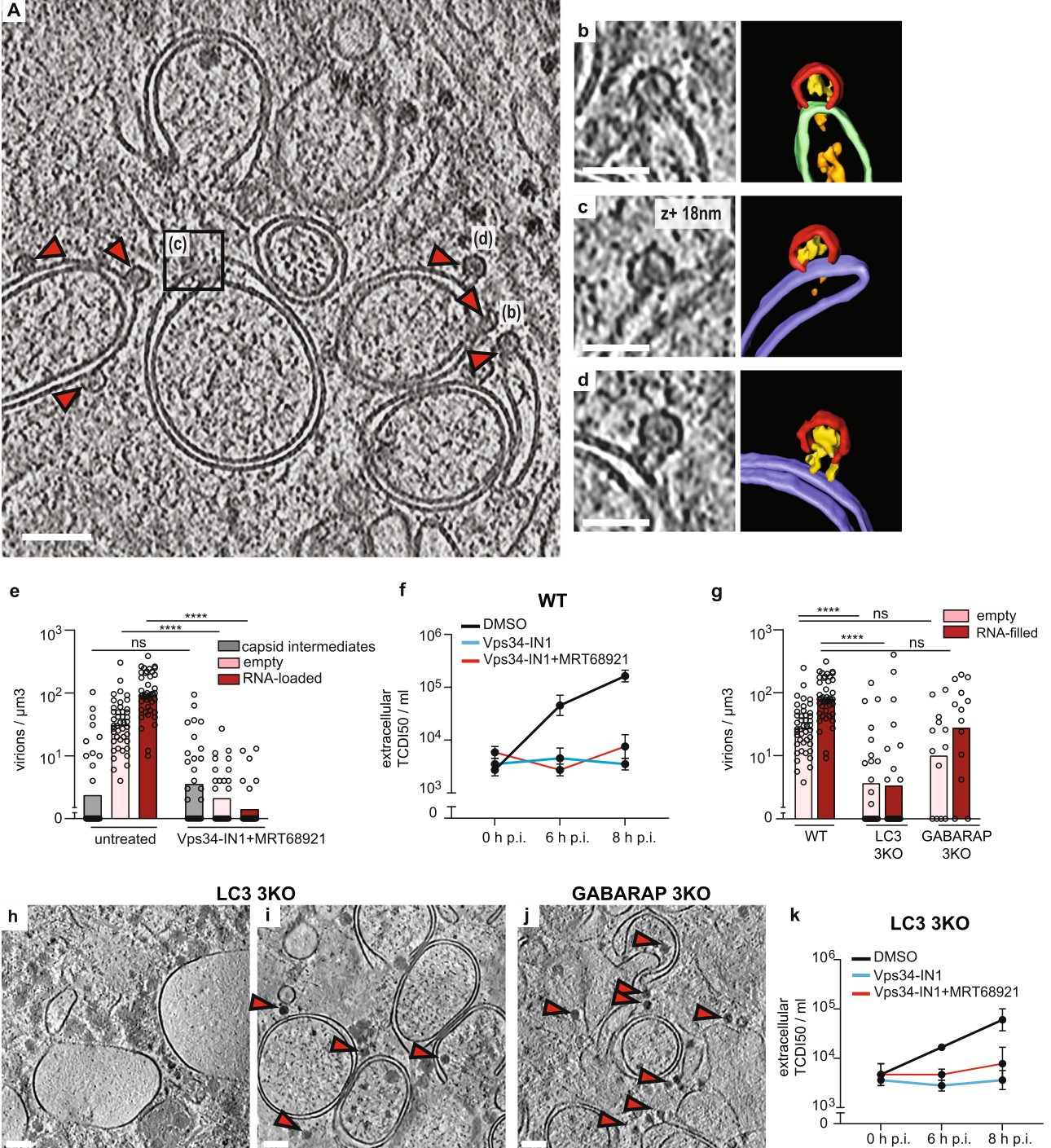

**Fig. 3 | VPS34 inhibition stalls PV assembly at the half-capsid intermediate.**
**a** Cryo-electron tomogram of a PV-infected cell co-treated with Vps34-IN1 and
MRT68921 at 6 h p.i. **b**–**d** Magnified views of capsid intermediates indicated in (**a**),
and their corresponding segmentations. Capsid intermediates (red), intra-capsid
densities (yellow), ALM (purple), SM (green), luminal densities (orange). **d** The
magnified view shows a near-complete capsid tethered to a double-membrane
vesicle. **e** Concentration of intracellular capsid intermediates, empty capsids and
RNA-loaded virions as observed in (*n*) tomograms of untreated (*n* = 51) and Vps34-
IN1 + MRT68921 (*n* = 32) treated cells at 6 h p.i., bars represent the averages (see
also Supplementary Table 2). Statistical significance by unpaired two-tailed Stu-
dent's *t* test; ****$p$ < 0.0001. **f** Released virus titer at 0, 6, and 8 h p.i. in WT cells
treated with DMSO, Vps34-IN1, and Vps34-IN1 + MRT68921. Bars represent the

means of biological triplicates ± SEM. **g** Concentration of intracellular empty cap-
sids and RNA-loaded virions observed in (*n*) tomograms WT (*n* = 51), LC3 (*n* = 24),
and GABARAP 3KO (*n* = 13) cells at 6 h p.i. Statistical significance by unpaired two-
tailed Student's *t* test; ****$p$ < 0.0001. **h**, **i** Slices through representative cryo-
electron tomograms of lamellas milled through PV-infected LC3 3KO cells at 6 h p.i.,
revealing two types of membrane proliferation: large single-membrane vesicles (**h**)
and ALM proliferation (**i**). **j** Slice of cryo-tomogram of a GABARAP 3KO PV-infected
cell at 6 h p.i., where ALMs were observed. (**h**–**j**) Red arrowheads indicate the
presence of RNA-loaded virions. **k** Time course of PV release from LC3 3KO cells in
the presence or absence of autophagy inhibitors as indicated in the figure. Error
bars represent the means of biological triplicates ± SEM. Scale bars: **a** 100 nm,
**b**–**d** 50 nm, **h**–**j** 200 nm.

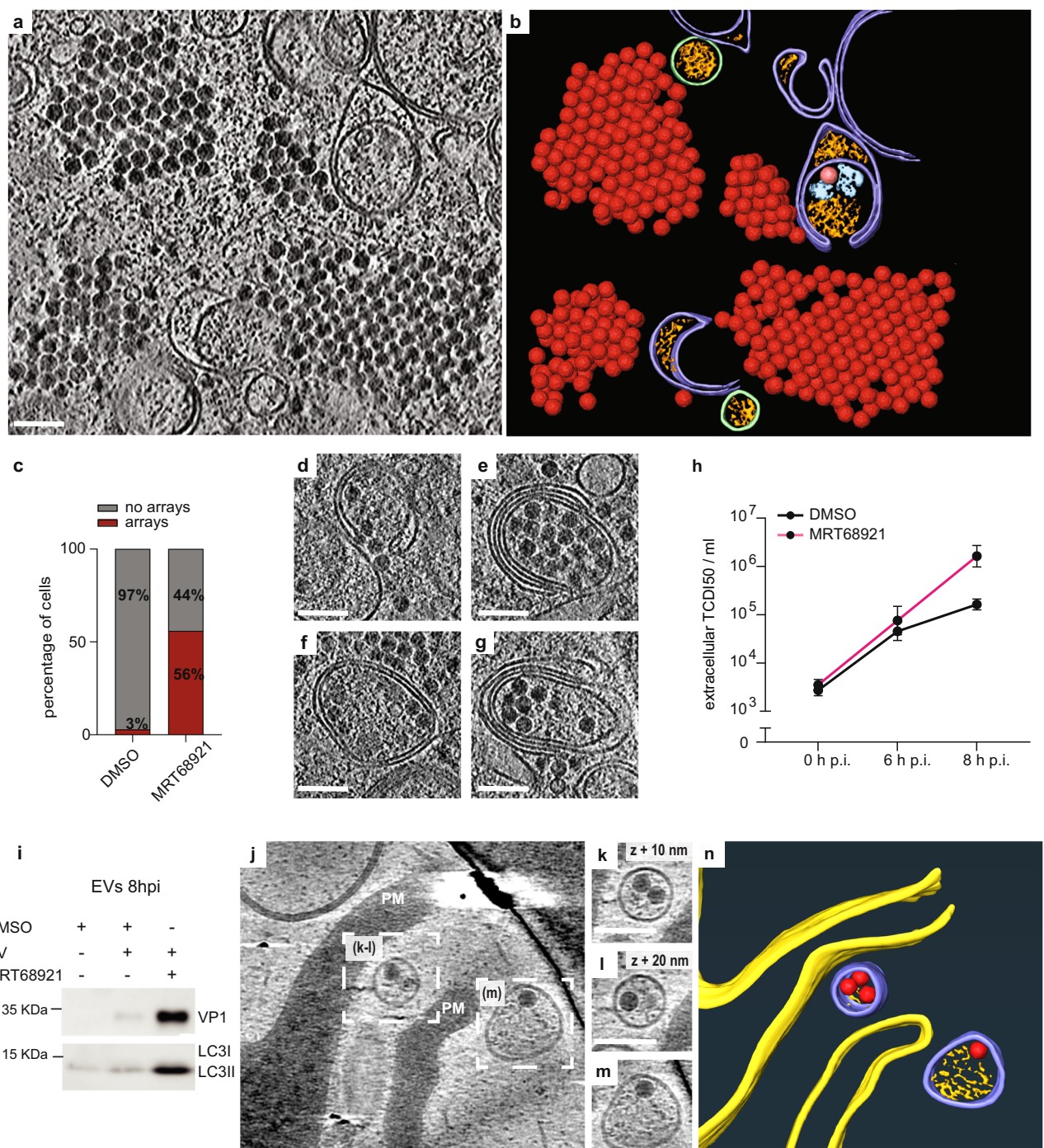

**Fig. 4 | Inhibition of ULK1/ULK2 induces formation of intracellular virus arrays and increases infectious virus release. a** Cryo-electron tomogram of an MRT68921-treated, PV-infected cell containing cytoplasmic virion arrays. **b** Segmentation of the tomogram in (**a**). Virions are represented by their sub-tomogram averages. Colors are as in Fig. 1b. **c** Percentage of cells containing virus arrays in DMSO and MRT68921-treated cells as measured from freeze-substituted sections. **d**–**g** Several examples of virion recruitment to ALMs in tomograms of MRT68921-treated, PV-infected cells. **h** Time course of PV release from MRT68921 and DMSO-treated cells. Error bars represent the means of biological triplicates ± SEM. **i** Western blot analysis of extracellular vesicles (EVs) harvested at 8 h p.i. from infected cells treated or not with MRT68921. **j** Slice through a cryo-electron tomogram recorded at the plasma membrane (PM) of a non-FIB-milled MRT68921-treated, PV-infected cell. **k**–**m** Magnified views of PV-containing released vesicles indicated in (**j**). **n** 3D segmentation of the tomogram in (**j**). PM (yellow), RNA-loaded virions (red), extracellular vesicles (purple), luminal densities (orange). Scale bars: 100 nm.

The tomograms allowed further classification of ALMs based on their contents (Fig. 5g–m). At 52%, the most abundant class was ALMs containing RNA-loaded virions (Fig. 5a–e, g). The second most abundant class, at 15%, was ALMs containing amorphous granular material that was clearly denser than the remaining cytoplasmic contents (Fig. 5g, h). These dense granules were frequently co-packed with virions (Fig. 5g, j). One distinct class representing 13% of ALMs contained tightly packed bundles of protein filaments (Fig. 5g, i, Supplementary Fig. 6a, b, Supplementary Movie 4). As opposed to the dense granules, filaments were rarely co-packaged with other ALM contents (Fig. 5g). The filaments were not affected by VPS34 inhibition, but instead completely absent in both LC3 and GABARAP 3KO cells

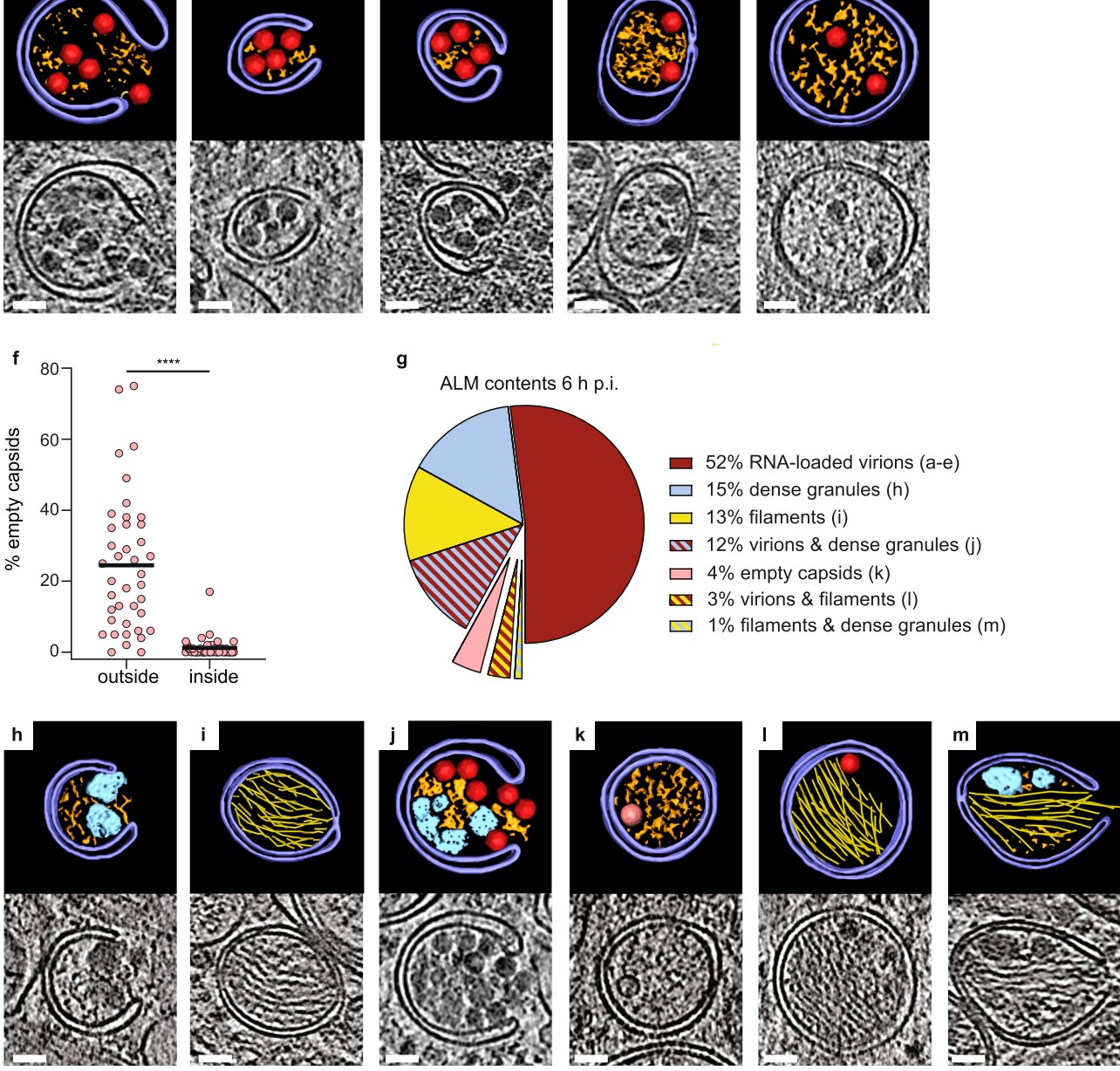

**Fig. 5 | Autophagy-like membranes select and sort their contents in PV-infected cells. a–e** Tomograms of PV-infected cells at 6 h p.i. showing different stages of engulfment of RNA-loaded virions by ALMs, including initial recruitment to phagophores (**a–c**) and enclosure in DMVs (**d**, **e**). Each panel contains a slice through the tomogram and the corresponding segmentation, colored as in Fig. 1b. **f** Percentage of empty capsids on the outside and inside of ALMs. Each dot corresponds to one tomogram analyzed ($n = 41$); horizontal line is the average (see also Supplementary Table 2). **g** Relative abundance of seven classes of ALMs by contents (single/mixed), in tomograms of PV-infected cells at 6 h p.i. **h–m** Segmentations and corresponding tomographic slices of examples of the different ALM classes, as labeled in (**g**). Colors are as in Fig. 1b. Protein filament bundles are shown in yellow. Statistical significance by unpaired two-tailed Student's $t$ test; ****$p < 0.0001$. Scale bars: 50 nm.

(Supplementary Fig. 6c). This is the opposite dependence on autophagy host factors than that displayed by virus assembly (Fig. 3g). We determined the filament structure by subtomogram averaging to 18.5 Å resolution, which yielded a helix with an average diameter of 10 nm, 29° twist and 52 Å rise per subunit (Supplementary Fig. 6d, e). From this map the identity of the filament could not be positively determined, but the map did allow definitive exclusion of protein filaments with known structure. A systematic comparison of the filament to all relevant mammalian protein filaments in the electron microscopy database allowed exclusion of all except decorated actin filaments (Supplementary Fig. 7 and Supplementary Table 3 for complete list). Thus, the filament is either actin decorated by a vinculin-like

actin-binding protein, or an unknown viral or cellular protein filament of similar size and shape.

Altogether, these data reveal an exquisite specificity in virus-induced autophagy: autophagy-like membranes selectively engulf RNA-loaded virions while excluding empty capsids, and co-package virions with dense granular material while segregating them from a second class of ALMs that contains bundles of protein filaments.

## Discussion
Here we present an in situ structural analysis of enterovirus replication by cryo-electron tomography. Our study focuses on the involvement of autophagy proteins and autophagy-like membranes in virion

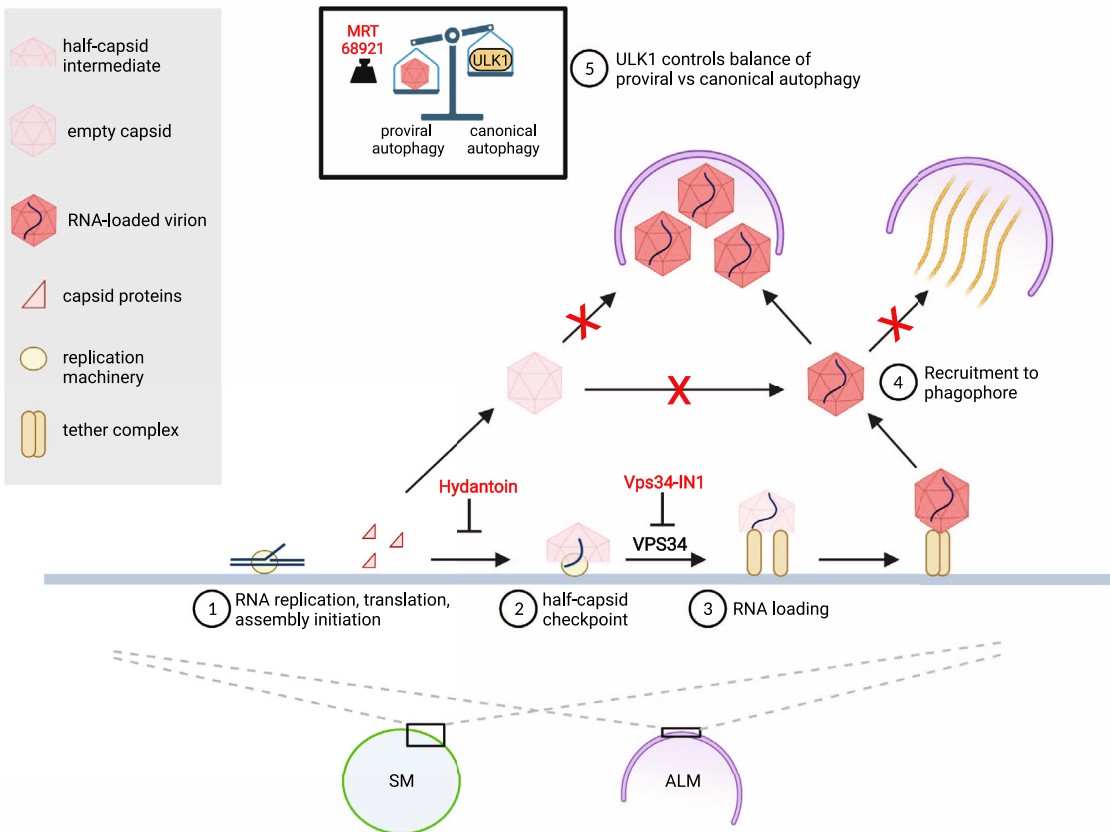

**Fig. 6 | The interplay between autophagy and enterovirus replication is dynamic, selective, and sequential.** This model illustrates how remodeled cytoplasmic membranes act as a platform for viral RNA replication and as a production line for virion assembly. (1) Capsid proteins, produced as part of the membrane-associated viral polyprotein, assemble on the replication membrane to the point of half-capsid intermediate. The antiviral drug Hydantoin leads to premature capsid release from membranes. Away from the replication membrane, capsids cannot be loaded with RNA. (2) The assembly pauses at the long-lived half-capsid intermediate that carries out the "late proofreading" resulting in exclusive incorporation of newly synthesized membrane-associated viral RNA[30]. The inhibition of VPS34 stalls the assembly at the half-capsid checkpoint, possibly by reducing RNA replication. (3) RNA-loading of the capsid leads to the formation of complete virions tethered to the replication membrane. (4) Phagophores selectively package RNA-loaded virions while excluding empty capsids. The virion-containing class of phagophores is distinct from a second class of phagophores containing bundles of protein filaments. (5) A balance between ULK1-induced canonical autophagy and virus-induced proviral autophagy regulates the level of virion production. Inhibition of remaining ULK1 activity by MRT68921 further increases virion production. Created with BioRender.com.

assembly and egress, and the data are consistent with the model presented in Fig. 6. Compared to viral RNA replication, much less is known about the site and pathway of enterovirus assembly. A membrane-proximal location of the assembly has been suggested, but direct evidence has been missing. In tomograms of infected cells, we identified an abundant capsid assembly intermediate structurally equivalent to half of an enterovirus capsid (Fig. 2). It was to 96% found directly docked to membranes in infected cells, and at roughly equal proportions on the cytosolic face of SMs and ALMs, which suggests that these two membrane types both serve as virus assembly platforms. Our findings are consistent with ultracentrifugation studies of enterovirus-infected cells that detected an abundant capsid-related species equivalent to half of an empty capsid, without being able to elucidate its identity[27,28].

The current paradigm of non-enveloped virus assembly holds that the proteinaceous capsid either assembles independently followed by energy-expending genome loading, or is templated by the genome[29]. Our finding extends that paradigm with a third mode: assembly that is assisted by the replication membrane and its bound components. A slow transition past the half-capsid intermediate may help ensure that the viral RNA has been incorporated before the capsid closes. This is consistent with a previously proposed "late proofreading" mechanism that aimed to explain why only recently produced viral RNA, present on the replication membrane, is incorporated into virions[30]. A direct structural coordination between the membrane-bound RNA replication and capsid assembly seems an appealing hypothesis but remains to be visualized in situ.

We showed that the lipid kinase VPS34 is a host factor required for assembly to progress past the half-capsid intermediate (Fig. 3), possibly due to its requirement for effective viral RNA replication (Supplementary Fig. 4h). The VPS34 requirement may explain why the membrane-bound picornaviral helicase 2C binds the class III PI3 kinase complex[31], in which VPS34 is the catalytic subunit. Hence, 2C may enable the virus to activate VPS34 independently of ULK1-dependent canonical autophagy induction. In fact, we show that pharmacological inhibition of ULK1 further boosts virus assembly and release (Fig. 4), and it was previously reported that poliovirus infection partially depletes ULK1[10]. The picture that emerges is that the virus optimizes the cellular environment by suppressing the master switch of canonical autophagy, ULK1, while at the same time activating necessary subsystems of the autophagy pathway, such as VPS34, in alternative ways. Notably, the class III PI3 kinase complex exists in two versions, with its fourth subunit being either ATG14 or UVRAG[32]. It remains to be seen if these two VPS34-containing complexes play differential roles in enterovirus assembly.

The cryo-electron tomograms revealed several layers of selectivity in enterovirus-induced autophagy (Fig. 5). Perhaps most remarkably, there is a strong selectivity for packaging of RNA-loaded virions

over empty capsids. Selectivity had not previously been demonstrated for virus-induced autophagy, but it is frequent in other forms of autophagy. There, so-called autophagy receptors mediate the degradation of specific cytoplasmic components by linking them to LC3 on the growing phagophore membrane[33]. The case of enterovirus particles thus seems to present a conundrum: How can phagophores selectively package RNA-loaded virions over empty capsids when the external surfaces of both these particles are virtually identical[34]? Further studies will be needed to elucidate this mechanism, but a clue may be provided by the correlation between RNA loading of capsids and their membrane tethering, as well as the loss of tethering upon autophagic engulfment (Fig. 1).

The tomograms allowed further structural catalogization of ALM contents in infected cells (Fig. 5). Viruses were frequently co-packaged with dense granular material which has also been seen in tomograms of vesicles released from enterovirus-infected cells[35]. While the composition of these granules is unknown, their high electron density is compatible with them containing RNA, which could mean that non-encapsidated viral or cellular RNA is released in the same vesicles as virions. As opposed to the copackaging with dense granules, virions were markedly segregated from ALMs that contained bundles of protein filaments. The filaments had a structural signature compatible with decorated F-actin. This tentative identity of the filaments would tie together previous reports that the actin cytoskeleton largely disappears in both starved and enterovirus-infected cells[36,37]. In starved cells, this is linked to presence of F-actin in the lumen of LC3-positive membranes (where it was suggested to play a role in shaping the phagophore)[37]. It is thus possible that the filament-filled ALMs represent structural snapshots of this process.

In summary, our study of poliovirus-infected cells by cryo-electron tomography reveals membrane-assisted capsid assembly and a link between membrane tethering and RNA-loading of virions. It further shows the multi-faceted nature of autophagy in enterovirus-infected cells, balancing virion production and autophagic engulfment, making sure that only RNA-loaded virions are packaged in phagophores and segregating them from other types of autophagic cargoes.

# Methods

## Cell lines and cultures
HeLa cells were obtained from ATCC (# CRM-CCL-2™). HeLa LC3 and GABARAP 3KO cells were described previously[38]. All cell lines were grown in (D)MEM supplemented with 10% fetal bovine serum (FBS)/ 25 mM HEPES/GlutaMAX™/Penicillin-/Streptomycin (Gibco) and maintained at 37 °C in a 5% $CO_2$ atmosphere. Cells were regularly screened for the presence of mycoplasma infection.

## Antibodies and virus
Rabbit monoclonal anti-LC3B (D11, Cell Signaling—diluted 1:1000), Mouse monoclonal anti-β-actin (A2228, Sigma—diluted 1:10,000), Mouse monoclonal anti-VP1(clone B3/H1—diluted 1:500), Rabbit polyclonal anti-3D gifted by George Belov (UMD—diluted 1:500), Human anti-A12 gifted by Konstantin Chumakov (FDA—diluted 1:10,000), Rabbit monoclonal anti-P62 (SQSTM1) (PM045, MBL—diluted 1/1000). Goat anti-Rabbit IgG (H + L) Cross-Adsorbed Secondary Antibody, Alexa Fluor 555 (A21428, Invitrogen—diluted 1:1000) and Goat anti-Human IgG (H + L) Cross-Adsorbed Secondary Antibody, Alexa Fluor 488 (A11013, Invitrogen—diluted 1:1000). HRP Goat Anti-Rabbit IgG H&L (ab205718—diluted 1:10,000) and HRP Mouse Rabbit Anti-Mouse IgG H&L (ab6728—diluted 1:10,000). Poliovirus Type 1 Mahoney strain was a gift from George Belov (UMD).

## Drug treatments
HeLa cells were pre-treated with DMSO or 1 μM MRT68921 (SML1644, Merck) in culture medium, 1 h before the infection. Medium was replaced with serum-free media containing DMSO or 1 μM MRT68921 and cells were infected with Poliovirus (PV) at MOI 5 (MOI 1 and MOI 0,1 for qPCR) for 1 h. Inoculum was then removed, and cells were incubated in fresh media containing 2% FBS and 1 μM MRT68921 and/or 5 μM Vps34-IN1 or 50 μg/mL hydantoin (5-(3,4-dichlorophenyl)-5-methylimidazolidine-2,4-dione (EN300-21815)), or 2 mM Guanidine hydrochloride (Sigma) for 3 h to 8 h. Vps34-IN1(17392, Cayman chemicals) was added to the cell media at 1 h p.i. to not interfere with the endocytosis-based viral entry. MRT68921 was added at 1 h p.i. After addition, all inhibitors were kept in the cell media throughout the experiments.

## LC3 lipidation assay
For each condition and time point, HeLa cells were seeded in two T25 flasks ($2 \times 10^6$ cells/flask) and infected with poliovirus at MOI 5. Infected cells were collected by trypsinization, then centrifuged at $500 \times g$ for 5 min, pellets were resuspended in PBS and washed twice with PBS. Cells were lysed with lysis buffer (20 mM Tris-HCl pH8, 300 mM KCl, 10% glycerol, 0.25% Nonidet P-40, 0.5 mM EDTA, 0.5 mM EGTA) on ice for 15 min, passed through a 22G needle and centrifuged at $21,000 \times g$ for 20 min. After centrifugation, supernatants were stored at −80 °C and protein concentrations were further determined using the Pierce™ BCA Protein Assay Kit (23225, ThermoFisher Scientific).

## SDS-PAGE/western blot
Proteins were separated by SDS-PAGE and gels were transferred via a semi-dry blotter to PVDF transfer membranes and blocked for 1 h with TBS-T containing 5% (w/v) milk powder or 5% (w/v) BSA, followed by probing with primary antibodies and overnight incubation, and further re-probed with corresponding HRP conjugated secondary antibodies, for 1 h. Blots were developed using SuperSignal West Pico PLUS Chemiluminescent Substrate (Thermo Scientific) and imaged using the Amersham Imager 600 (GE Healthcare Biosciences) or the Biorad ChemiDoc™ Touch, and analyzed with imageJ. Uncropped western blots are available in data source and supplementary information files.

## Sample preparation for cryo-electron tomography
24 h before infection, cells were seeded on R2/2 gold UltrAufoil grids (200 mesh, Quantifoil Micro Tools GmbH, Großlöbichau, Germany) in μ-Slide 8 Well chamber (IBIDI) at $2 \times 10^4$ cells/well. Prior to use, UltrAufoil grids were glow-discharged, dipped in ethanol, and then washed with cell media for 30 min. HeLa cells were infected with poliovirus at MOI 5 for 1 h in serum-free media at 37 °C. The μ-Slide was gently agitated every 15 min to ensure an even coverage and maximize virus contact with the cell monolayer. After 1 h of virus absorption, the inoculum was replaced with fresh DMEM media supplemented with 2% FBS. Infected cells were plunge-frozen in liquid ethane-propane mix 3 h or 6 h post-infection using a Vitrobot plunge freezer (Thermo Fisher Scientific) at 23 °C, 90% humidity, with blot force −5 and blot time 6.5 s.

## Cryo-lamella generation and characterization
Cryo-lamellas of poliovirus-infected cells were generated employing the wedge-milling method[39] using a Scios focused-ion-beam scanning electron microscope (ThermoFisher Scientific). To prevent sample drift during the milling process and to enhance sample conductivity, the samples were first coated with platinum using the gas injection system (GIS, ThermoFisher Scientific) operated at 26 °C, at 12 mm stage working distance and 7 s gas injection time. The milling was performed at a tilt angle within a range of 17°–23° stage tilt. Lamella preparation was performed in a stepwise milling using parallel rectangular pattern above and below the area of interest, with reducing the ion beam current throughout the milling process, from 0.3 nA for the first milling step to remove the bulk material to 0.03 nA at the final cleaning

step to obtain the lamella which was set to a minimum thickness of 200 nm. To minimize the contamination of lamellas with ice crystals, they were stored in liquid nitrogen for less than a week before tilt-series collection at the Titan Krios (ThermoFisher Scientific). The final thickness of lamellas was measured at the Titan Krios. Two images of the same area of a lamella were recorded at an intermediate magnification (×8700): an energy-filtered image (F image), and non-filtered image for which the energy filter slit was removed (nF image). The lamella thickness was estimated as 350*ln(I(nF)/I(F)), where I(nF) and I(F) are the intensities in the non-filtered and filtered images, respectively, and 350 the estimate of the electron mean-free path at 300 kV in ice (in nm).

### Cryo-electron tomography

Data was collected using a Titan Krios (ThermoFisher Scientific) operated at 300 kV in parallel illumination mode. Tilt series were recorded using SerialEM software on a K2 Summit detector (Gatan, Pleasanton, CA) operated in super-resolution mode. The K2 Summit detector was mounted on a BioQuantum energy filter (Gatan, Pleasanton, CA) operated with a 20 eV slit width. A condenser aperture of 70 μm and an objective aperture of 100 μm were chosen for tilt-series collection. Coma-free alignment was performed with AutoCtf/Sherpa (ThermoFisher Scientific). Tilt-series were acquired in dose-symmetric or bi-directional mode. Due to the pre-tilt of the lamellas, the starting angle used was +13° for dose-symmetric, and −21° to −25° for bi-directional tilt series acquisition. The following parameters were used for acquisition: 33 kx nominal magnification corresponding to a specimen pixel size of 2.145 Å; defocus range −3 to −5 μm, tilt-range depending on the lamella pre-tilt and thickness typically ±50° to ±60°; tilt increment 2° or 3°; total electron dose ~100 e⁻/Å² (bi-directional tilt series) and 130 e⁻/Å² (dose-symmetric tilt series). The exposure dose was not varied as a function of tilt angle. At each tilt angle, the exposure was saved as a non-gain-corrected TIFF movie containing a dose per frame of around 0.25 per super-resolution pixel.

### Cryo-electron tomography data processing

Super-resolution TIFF movies were unpacked and gain-reference corrected, and subsequently corrected for sample motion using MotionCor2[40]. The motion correction included a factor 2 binning resulting in a specimen pixel size of 4.29 Å. After reassembly of tilt-series image stacks they were processed using IMOD[41]. Tilt series were aligned using patch tracking. The aligned stacks were CTF-corrected with a custom-made script using CTFFIND4[42] and CTFPHASEFLIP[43], a part of IMOD[41]. Tomograms were then generated in IMOD using weighted back projection, no low-pass filtering was performed at this stage. For visualization, tomograms were 3 times binned using IMOD, resulting in a pixel size of 12.87 Å. The isotropic resolution and signal-to-noise of the tomograms were improved using the deep-learning software IsoNet software[44]. Filtered tomograms were further segmented using Amira for the representation of membranes, protein densities and filaments, the last using the Fiber tracing module. Subtomogram averages of empty capsids and RNA-loaded virions were integrated in these 3D renderings through UCSF Chimera[45].

### Subtomogram averaging of viruses

Subtomogram averaging of tethered virions and virus particles was performed using Dynamo[46,47]. 241 tethered virions were manually picked from several tomograms (box size 180*180*180 voxels), of these particles, 179 could be clearly oriented and centered manually. The orientated particles were further used for single-class alignment with spherical mask with radius of 80 px, first allowing shifts only (Supplementary Fig. 8A). A second round of alignment was performed allowing full azimuthal rotations and 30° tilt. The resolution of the final

average was calculated to 70 Å using the Gold-standard Fourier Shell Correlation with a threshold of 0.143.

For RNA-loaded virions, a total of 566 particles were manually picked in several tomograms (Supplementary Fig. 8B). For empty capsids, a total 426 particles were manually picked (Supplementary Fig. 8C). Subvolumes of unbinned particles (4.29 Å/px) were extracted with a box size of 140*140*140 voxels. Subvolumes were first centered on their average allowing only shifts using a spherical mask with radius of 60 px and a Gaussian falloff of 3 px. After centering, two rounds of a five-iteration single-class alignment were performed, each with icosahedral symmetry imposed and rotational alignment allowing for full rotations within one symmetric unit (+/−45°). Before the second round, starting angles were randomized. The subtomograms were then subjected to a four-class multireference alignment (MRA) in Dynamo. Two of the five classes for both empty and RNA-loaded virions had a similar appearance, and were pooled for a second round of Gold-standard single-class alignment with imposed icosahedral symmetry. Gold-standard FSC curves for resolution estimation of the virus averages were calculated in Dynamo, resulting in a resolution estimate of 21.4 Å for empty capsids and 28.0 Å for RNA-loaded virions at a Fourier shell correlation threshold of 0.143. Subtomogram averages were low-pass filtered to their respective resolution and the 3D renderings were created in UCSF Chimera.

### Subtomogram averaging of filaments

The subtomogram selection, extraction and averaging followed the workflow schematically presented in Supplementary Fig. 9. 1856 filaments were traced in 16 tomograms using the fiber tracing module[48] in Amira (Thermo Fisher Scientific). The tracing was performed within manually segmented regions corresponding to filament-filled autophagy-like membranes, and the tracing parameters were selected so that the number and length of the filaments was consistent with their visual appearance in the tomograms. Filament coordinates were exported from Amira and imported to Dynamo using a custom-written MATLAB (Mathworks) script (available at https://github.com/Lars-AndersCarlson/Filament), after which Dynamo was used to extract subtomograms at regular intervals. Initially, an oversampled subtomogram extraction was performed, with (100 px)³ subtomograms extracted along the filament axis at 5 px intervals. This resulted in 55,376 subtomograms. Each subtomogram was assigned initial Euler angles that aligned it with the traced filament axis, but gave it a randomized rotation along that axis. The subsequent alignments and classifications were performed using a cylindrical mask with a radius of 16 px and a Gaussian falloff of 3 or 5 px. Subtomogram averaging was performed in Dynamo. After an initial single iteration allowing for centering of the subtomograms perpendicularly to the filament axis, several iterations were performed allowing for shifts, free rotations around the filament axis, and a +/−30° tilt with respect to the filament axis. At this stage of the data processing, efforts to determine filament polarity by allowing each subtomogram in a given filament to change orientation during alignment, and then imposing the majority orientation on all subunits of the filament were not yet successful. From the full oversampled dataset, overlapping subtomograms were removed using a distance threshold of 10 px (42.9 Å), resulting in a set of 16,682 subtomograms. These subtomograms were subjected to a five-class multireference alignment (MRA) in Dynamo. Three of the five classes had a similar, regular helical appearance, and were pooled for a second round of MRA with five classes, from which one class of 3517 subtomograms was better defined than the others. This class was subjected to a half-set gold-standard refinement in Dynamo resulting in a resolution estimate of 23.8 Å at a Fourier shell correlation threshold of 0.143. The resulting average allowed a first estimation of the helical parameters as a rise of ~56 Å per subunit and a rotation of ~36° per subunit. At this point, tomograms were re-reconstructed

using NovaCTF to allow for 3D CTF correction by phase flipping[49]. Subtomograms were reextracted at positions corresponding to the helical subunits, which after removal of overlapping particles resulted in an enlarged data set of 10897 subtomograms. These subtomogram poses and positions were exported from Dynamo for further processing in the subTOM package written in MATLAB with functions adapted from the TOM[50], AV3[51], and Dynamo packages. The scripts and relevant documentation are available to download [https://www2.mrc-lmb.cam.ac.uk/groups/briggs/resources]. In addition, instead of a binary wedge mask, a modified wedge mask was used[52]. The missing wedge was modeled at all processing stages as the average of the amplitude spectra of subtomograms extracted from regions of each tomogram containing empty ice, and was applied during alignment and averaging. We applied both principal component analysis (PCA) and multivariate statistical analysis (MSA) to classify subtomograms by filament straightness and similar helical parameters. The PCA was performed on wedge-masked difference (WMD) maps[53] with calculations implemented in MATLAB using code adapted from PEET[53] and Dynamo packages, and MSA also implemented in MATLAB using code adapted from I3[54]. Movies detailing the variability related to each Eigenvolume from MSA classification were generated using Eigen-filtering/reconstruction methods, and similar movies from WMD classifications were generated by producing class averages sorted by determined Eigencoefficients. Improved class averages allowed for recalculation of helical parameters and further sub-boxing as well as determining the filament polarity. Specifically, the sub-box poses and positions were determined by auto-correlation of the reference after rotation along the filament axis, which yielded a helical symmetry with a rise of 51.5 Å and rotation of 29°, and then duplicates within a sphere of diameter 50 Å were removed. Poses were not adjusted from their gold-standard alignment values and the Fourier shell correlation determined resolution was improved to 18.5 Å at the 0.143 threshold. Following the FSC calculation, the half-maps were averaged, filtered to the measured resolution by the determined FSC-curve and sharpened using a heuristically determined B-factor of $-4000$ Å$^{2,55}$.

### Quantitative analysis of membrane structures and virions

Tomograms were visually inspected and membrane structures such as single-membrane vesicles and tubes were assigned as single membranes. Double membrane structures were assigned as phagophore-like membranes if they had a clear opening within the tomogram volume (i.e., were cup-shaped), and as double-membrane vesicles if they did not have an opening. Collectively, these two types of double-membrane structures were referred to as autophagy-like membranes. Virus particles were localized and counted in each tomogram with template matching using PyTom[56]. For empty capsids and RNA-loaded virions template matching was performed using respectively the re-sampled, filtered empty capsid (EMD-9644) and mature virion (EMD-9642) cryo-EM structures of the human coxsackievirus A10[57]. The concentration of each structures was obtained by dividing the total number of structures by the volume of the corresponding tomogram. The number of tomograms obtained for all the conditions are listed in Supplementary Table 1. The closure of viral capsid intermediates was calculated using IMOD. First, in a central section through the capsid intermediate, the circumference was traced and measured with IMOD drawing tools, then the open length was divided by the circumference of a complete virus particle. This fraction was multiplied by 360 to get the angle of the cone that describes the capsid assembly intermediate.

### Freeze substitution

Poliovirus-infected HeLa cells non-treated and treated with MRT68921 were grown on carbon-coated sapphire discs and high-pressure frozen at 6 h p.i. using a Leica HPM100. Freeze substitution was performed as a variation of the Kukulski protocol[58]. Briefly, sapphire discs placed in adapted carriers were filled with freeze substituent (0.1% uranyl acetate in acetone, 1% H$_2$O) and placed in a temperature-controlling AFS2 (Leica) equipped with an FPS robot. In the first step, freeze-substitution occurred at $-90$ °C for 48 h then the temperature was raised to $-45$ °C. The samples were maintained in the freeze substituent at $-45$ °C for 5 h before washing 3 times with acetone followed by a temperature increase and infiltration with increasing concentrations of Lowicryl HM20. Finally, the samples were gradually warmed up to $-25$ °C before infiltrating 3 times with 100% Lowicryl and UV-polymerized for 48 h at $-25$ °C. Polymerization then continued for another 24 h at room temperature. The embedded samples were sectioned with a diamond knife (DiATOME-90) to slices of 60–120 nm thickness by using ultra-microtome (Reichart ULTRACUT S). Imaging was performed in FEI Talos electron microscope operating at 120 kV. Grids were examined at a Talos L120C (FEI, Eindhoven, The Netherlands) operating at 120 kV. Micrographs were acquired with a Ceta 16M CCD camera (FEI, Eindhoven, The Netherlands) using TEM Image & Analysis software ver. 4.17 (FEI, Eindhoven, The Netherlands).

### Confocal microscopy analysis

HeLa wild-type (WT), LC3 3KO, and GAB 3KO cells were infected with PV for 1 h at MOI 5 in serum-free media, washed and kept in 2% FCS DMEM/high glucose for 3 h or 6 h. Cells were fixed in 4% paraformaldehyde (PFA)/phosphate buffer solution (PBS) for 10 min at room temperature. Primary and secondary antibody incubations were carried out in PBS/10%FBS supplemented with saponin at 0.2% for 1 h at room temperature. Cells were rinsed twice in PBS, twice in water and mounted with Dapi Fluoromount-G (Electron Microscopy Science). Image acquisition was performed using a LSM780 confocal microscope (Carl Zeiss) with a 63X/1.4 NA oil objective or a 40X/1.4 NA oil objective. Cell numbers and individual cells respective mean fluorescence intensities and area were obtained using ImageJ.

### Virus titration analysis of cell supernatants

Extracellular medium was collected from infected cell cultures and serially diluted in 96-well plates ($10^{-1}$ to $10^{-8}$). Dilutions were subsequently used to inoculate in triplicates HeLa WT cells seeded at $6 \times 10^4$ cells/well in 96-well plates. Cells were incubated at 37 °C for 44 h, fixed with 10% paraformaldehyde, and stained with crystal violet. TCID50/ml was calculated using the Spearman & Kärber algorithm.

### Vesicle isolation

HeLa wild-type (WT), LC3 3KO, and GAB 3KO cells were infected with PV for 1 h at MOI 5, washed, and kept in serum-free DMEM/high glucose for 8 h. Supernatants obtained from one well of a 6-well plate (1.5 mL, ~500,000 cells) were harvested and centrifuged at $1000 \times g$ for 10 min, then at $20,000 \times g$ for 30 min. Pellets were either resuspended in 1X loading buffer and analyzed by western blot, or resuspended in RNA lysis buffer and processed for RT-PCR.

### Quantitative (q) PCR analysis

Cell supernatants and cell lysates were harvested at specific time points and lysed using RNA lysis buffer provided in the RNA isolation kit (Quick-RNA Microprep Kit, Zymo Research). RNA isolation was performed as per the manufacturer's instructions and cDNA was prepared using Thermo Scientific Maxima First Strand cDNA Synthesis Kit for RT-qPCR (Fisher Scientific). RT-PCR was performed using iTaq Universal SYBR® Green Supermix (BioRad) in Roche LightCycler 96 system (Roche), using the following thermal cycling conditions: 95 °C for 90 s, 40 cycles at 95 °C for 10 s, 57 °C for 10 s, and 72 °C for 110 s. The samples were run in duplicate for each data point. Primers used: For 5'-CGGCTAATCCCAACCTCG-3', Rev 5'-CACCATAAGCAGCC ACAATAAAATAA-3'. The genome copy numbers were standardized using a series of 10-fold dilutions ($10^3$–$10^{10}$ copies mL-1) of synthetic cDNA oligos: 5'-CCCCTGAATGCGGCTAATCCCAACCTCGGAGCAGGT

GGTCACAAACCAGTGATTGGCCTGTCGTAACGCGCAAGTCCGTGGCG
GAACCGACTACTTTGGGTGTCCGTGTTTCCTTTTATTTTATTGTGGCT
GCTTATGGTGACAATCACAG -3' (gBlocks® Gene Fragments, IDT).

### Statistics and reproducibility
Data and statistical analysis were performed using Prism (GraphPad Software Inc., USA). Details about replicates, statistical test used, exact values of $n$, what $n$ represents, and dispersion and precision measures used can be found in figures and corresponding figure legends. Values of $p < 0.05$ were considered significant. All tomograms shown are representative of larger data sets as indicated in Supplementary Table 2.

### Reporting summary
Further information on research design is available in the Nature Research Reporting Summary linked to this article.

### Data availability
The subtomogram average maps have been deposited in the Electron Microscopy Data Bank (EMDB) under accession codes EMD-15390 (empty capsid), EMD-15391 (RNA-loaded virion), EMD-15392 (tethered virion), and EMD-13682 (filament). Source data are provided with this paper.

### Code availability
The script used to pick filament subtomograms from Amira filament tracings is available at https://github.com/Lars-AndersCarlson/Filament.

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

## Acknowledgements

We would like to thank Carlson lab members, Sven Carlsson, and Richard Lundmark for constructive suggestions, Dale Corkery for help with the LC3 lipidation assay, as well as Nitesh Mistry for help with virus purification (all Umeå University). We are also thankful to Sebastian Schultz and Andreas Brech (Oslo University) for sharing their experience in freeze substitution. Cryo-EM data were collected at the Umeå Centre for Electron Microscopy (UCEM), a SciLifeLab National Cryo-EM facility and part of National Microscopy Infrastructure, NMI VR-RFI 2016-00968), supported by instrumentation grants from the Knut and Alice Wallenberg Foundation and the Kempe Foundations. This project was funded by a postdoctoral grant to S.D. from the European Union's Horizon 2020 research and innovation program under the Marie Skłodowska-Curie grant agreement No 795892. Additional funding came from the Human Frontier Science Program (Career Development Award CDA00047/2017-C to L.-A.C.), the Knut and Alice Wallenberg Foundation (through the Wallenberg Centre for Molecular Medicine Umeå). N.A.-B. and A.K. were intramurally funded by the National Institutes of Health (USA).

## Author contributions

S.D., A.K., K.S., N.A.-B., and L.-A.C conceived the project. S.D. performed cryo-electron tomography. A.K. performed virus replication and egress assays. S.D., D.R.M., and L.-A.C. performed subtomogram averaging. S.D., A.K., K.S., B.A., D.R.M., N.A.-B., and L.-A.C. analyzed data. S.D. and L.-A.C. wrote the original draft. All authors reviewed and edited the manuscript.

## Funding

## Competing interests

The authors declare no competing interests.
