## [Peer Review File · Nature Communications]

Reviewer Comments, first round

Reviewer #1 (Remarks to the Author):

In this manuscript, Dahmane et al. provide the first analysis of enterovirus replication structures by in situ cryo-electron tomography. The authors convincingly show that these virus-induced structures are also involved in enterovirus assembly, and that there is some selective packaging of RNA-loaded virions into the virus-induced double-membrane vesicles (DMVs, in this study termed autophagy-like membranes, ALMs). Moreover, the authors also investigate the role of the autophagy players and establish that, while canonical autophagy may not be crucial, the autophagy factor VPS34 facilitates virus encapsidation, whereas ULK1 may counteract virus assembly and release. The data analysis is sound, the images are breathtaking and show an exquisite level of detail. Although many questions remain open at a mechanistic level and even the identity of some newly-visualized densities is unclear, in my view, this is a wonderful example of how this novel methodology can provide new insight on viral infection, naturally opening a variety of follow-up questions.

Specific comments:

- I find the use of the term autophagy often rather confusing, starting with the title. It is fine to call the virus-induced double-membrane vesicles ALMs due to their morphological resemblance with autophagosomes, but these appear to be quite different functionally (not for degradation but, at least in part, for the release of virus particles). Moreover, as stressed in this study, it does not seem that canonical autophagy plays a specific role in the formation of ALMs. It would be good to find alternative formulations that make this distinction clear. For example, in the title, "selective autophagy of enteroviruses" is very unclear ("selective engulfment in ALMs"?). Other examples of confusing terms: "virus-induced autophagy", "autophagic membranes l. 182", "phagophores" l. 187, 254-5, "autophagic engulfment" l. 274). It would be good to thoroughly revise the text with this in mind.
- Although some EM studies on enterovirus-remodelled membranes are cited, the study on poliovirus (Belov et al. PMID 22072780) is inexplicably missing; please, add. This study (and reference 11) provided data supporting that ALMs arise from the single membrane structures. In the article, this is not mentioned and both types of virus-induced structures appear as completely independent. Importantly, this may have implications for the selective engulfment of RNA-loaded viral capsids into DMVs/ALMs. This could well occur concomitantly with the single to double membrane transformation, instead of the totally independent recruitment that the text and the model in Fig. 6 suggest, which, indeed presents some conundrums (l. 255).
- Enterovirus viral RNA synthesis is associated with both single-membrane structures and ALMs (Belov et al., Melia et al. PMID 29045829) and not only with single membrane structures as l. 49 suggests. This is perhaps not surprising considering that one type seems to arise from the other, as mentioned in the previous point. Nicely, the current data show that encapsidation, likely linked to replication (ref. 28), can occur on both types of membrane structures.
- VPS34 is a subunit of the class III PI3K complex formed in the initial steps of autophagy, but plays other roles, for example, in the endocytic pathway. Inhibiting other subunits of the class III PI3K complex would be a great addition, but considering the time-consuming cryo-tomography workflow, this would be understandably outside the scope of this work. However, these are points worth including in the discussion. Along the same lines, there is no direct evidence in this study that virus-induced structures are PI3P-decorated (l. 149), so it would be appropriate to tone down this statement.
- L. 79. "Requires" appears too strong. The formulation of this point later in the text (l. 109) appears more precise. Even though the data is highly suggestive, as far as I can tell, it cannot be formally excluded that alternative, non-membrane associated, encapsidation paths co-exist. Although this is not explicitly mentioned, I assume that regions containing virus-induced

membrane structures were mainly (if not only) selected for data collection, and therefore, not only the sampling is not random, but other regions of the cytoplasm have not been analysed or not in so much detail.

- Old studies (e.g. Belov et al & ref. 11), also reported the presence of multilamellar structures, which appeared to be further enwrapped ALMs. Were such structures completely absent in the samples? I think I see a glimpse of one in Figure S1.
- The legend of supplementary figure S3 appears to be missing in the uploaded submission.

Reviewer #2 (Remarks to the Author):

In the present manuscript, Dahmane and co-workers describe features of poliovirus assembly in infected cells by using a cryo-electron tomography-based approach. The authors provide compelling evidence that poliovirus capsids assemble predominantly on the cytoplasmic surface of SMs and ALMs, assumed to carry the viral replication machinery. Among the findings, the authors identify a prominent half-capsid assembly intermediate, progression from which to complete virus particles depends on the class III phosphatidylinositol 3-kinase (Vps34), but not the canonical autophagy inducer ULK1/2. Interestingly, inhibiting ULK1/2 strongly increases the appearance of arrays of assembled poliovirus particles. On the cryo-EM part, the authors acquired ~200 tomograms and complemented the qualitative analysis with subtomogram averaging of viral capsids and filaments in the context of virus-induced vesicles.

Overall, the study addresses an interesting and novel aspect of virology, with interest for a broad readership. To the most part, the study has been performed well and with great care. However, several parts require important additional experimentation to ascertain the main findings of the study.

The major concerns are:

1. A major limitation of this study is the lack of detection of viral RNA and the viral replicase machinery. It is claimed that newly formed viral RNA is packaged into capsids. However, there is no data on detection of viral RNA (e.g. by using FISH-based approach) and the viral replicase (e.g. by super-resolution microscopy or immunolabelling of the RdRp or other component of the replicase). One would assume that the replicase sits close to the site where viral RNA is transferred to the half capsids. Alternatively, viral RNA is amplified somewhere else on the membrane surface and then transported to the site of capsid formation. From my point of view, this should be addressed to support the authors claims and the model.
2. Figure 4 A-G: There is no information about the tethering of putative PV particles in the context of ULK-inhibition. In the images provided, it seems that no tethered particles are found, but given the increase in infectivity, the assembly of the particles should also increase. Is the absence (lack of detection) of tethered particles due to very much accelerated assembly?
3. Line 116: the authors mention variable levels of densities in the intermediate capsid structures (Fig 2B-E). At the shown resolution, it is difficult to discern if these densities are different or are RNA. The authors should validate in an alternative way whether or not the lumen of assembled capsids contains viral RNA. This is also the case for Figure 4I: The authors could prove via immunolabelling or negative staining that the electron-dense round structures are indeed PVs inside the extracellular vesicles or viral RNA inside the newly encapsidated structures.
4. Figure 1, Figure 5: The authors claim assembly occurs on autophagy membranes but there is no marker for this. Perhaps it would be possible to use the same Kukulski protocol used for the thin sections analysis to perform CLEM with a fluorescent marker for autophagy? In addition, the authors could prove via immunolabelling or negative staining that the electron-dense round structures are indeed PVs inside the extracellular vesicles.
5. Figure 3F: The author should include MRT68921 treatment only and also measure the specific infectivity of extracellular viruses (e.g measure the amount of extracellular viral capsid, viral RNA) to clarify the decrease in virus release or virus infectivity in VPS34-treated cells.
6. Line 138: The authors claim that VPS34 inhibition does not affect viral RNA replication.

However, Figure S4G shows some reduction in intracellular viral RNA in VSP34 treated cells although it is not statistically significant. In addition, by admission of the authors VPS34 is required for high level membrane proliferation. A proper discussion of this observation is required. 7. Into the same vein, please include MRT68921 treatment only and also a positive control (e.g. guanidine-HCl). In this experiment, different MOIs should also be tested (e.g. 1, 0.1...).

Minor points:

1. The authors state the membrane tethering complex is regular in length. Did the authors consider trying a further characterization, perhaps by subtomogram averaging?
2. Line 157 typo, inhibition with capital I
3. Line 421: Tomogram averaging of empty and full capsids: Is there any structural difference in the capsid, except RNA loading? Did the authors observe the icosahedral structure of PV capsid in any of the assembled structures?
4. Line 490: Please explain why you used coxsackievirus for template matching rather than the structure calculated by the virion arrays. Is there any difference between the published structure and the one determined by subtomogram averaging?
5. Line 794: wrong colour code for ALM (purple). It should be blue.
6. Line 838: typo "tomigram"
7. Line 421: Tomogram averaging of empty and full capsids: Is there any structural difference in the capsid, except RNA loading? The icosahedral structure could be shown? Are empty capsids known to have some biological function or are they just a byproduct? Please comment.
8. Fig 3F: If vesicular release is affected in ULK1/2 inhibited cells, intracellular infectivity should also be higher. Did the authors measure this?
9. Figure 5: EM images have very strong contrast and very little grey level. It would be helpful to see the images at a lower binning.
10. Table S2, line 568: numbers seem to be in contradiction of the data shown in Figure 1 plot E and K. Perhaps rows 1-3 were inverted?
11. Figure S4A. The author should include VPS34 treatment only and include other markers of autophagy e.g. p62. The electron-dense dots budded from the SMs and ALMs, and in the vicinity of these membranes (Figure 1), were identified as PV particles. This observation was confirmed by Hydantoin treatment to prevent RNA loading of PV capsids. However, these budding "virions" are rather reminiscent of spherules that are produced next to ALMs by MERS-CoV and infectious bronchitis virus (IBV). This point should be discussed.
12. Table S2, line 568: Numbers seem to be in contradiction of the data shown in Figure 1 plot E and K. Perhaps rows 1-3 were inverted?

Reviewer #3 (Remarks to the Author):

The manuscript by Dahmane and colleagues attacks, from a structural perspective, one of the most difficult questions in virology – how are nonenveloped RNA virions assembled in the cytoplasm of infected cells. While single particle cryo-electron microscopy has opened the gates for a deluge of structures describing various intermediates associated with viral replication, understanding the behaviour of these proteins in their cellular context is still a difficult task. The present study uses the laborious strategy of preparing FIB cryo lamellae from infected cells and employs cryo-electron tomography to reconstruct the three-dimensional landscape of specific cellular compartments.

The model of choice is poliovirus – the most studied member of enteroviruses – for which a large volume of biochemical information is available regarding the packaging of mature virions and their release either through the common lytic process or through the shedding of virions wrapped in cytoplasmic membranes.

The present article is a tour de force from a microscopic point of view and clarifies a number of long standing questions in the picornavirus field. First, the authors show that poliovirus capsids are packaged if and only if they are attached to cytoplasmic membranes. Secondly, they identify, in-situ, two important assembly intermediates: the first, corresponds to roughly a half capsid and is associated with genome packaging, while the second is tethered at a constant distance from the membranes. Both of these structures are present only on the cytosolic side of the membrane and are absent in the luminal space. Thirdly, they show that there is a strong selectivity for full virions in cytoplasmic vesicles. Finally, the authors find cellular factors involved in the progression over

the half capsid intermediate, factors regulating the release of cargos transporting virions via the autophagy pathway, and factors controlling the formation of intracellular viral arrays and virion release. The results are supported by solid statistics obtained from a reasonable number of tomograms obtained from cellular lamellae obtained under different conditions.

Overall, I consider that this manuscript will be of extraordinary importance for virologists and it will open a rich avenue for further studies. I warmly recommend its publication.

I have few minor points that, if addressed by authors, would improve the manuscript.

- The authors have performed subtomogram averaging of full and empty capsids imposing icosahedral symmetry. However, despite the larger number of full capsids the resolution attained was lower than for empty capsids. Did the authors try to see if more than one assembly intermediate can be identified in the population of full capsids? Do all capsids show the same amount of internal density? I suggest the authors to include a supplemental figure with the capsid analysis.

- The presence of the membrane associated half-capsid intermediate is a very intriguing finding of this study. While I realize that there is a small number of subtomogram volumes trapping this assembly stage, I wonder if the authors have tried to see if there is any symmetry related information in these particles. Is a 5-fold axis perpendicular to the membrane? Using the icosahedral reconstruction would be possible to obtain a missing wedge corrected cross correlation value for each of these volumes.

- The same question regarding the tethered capsids.

- Did the authors attempt the reconstruction of the tethering domain? While the resolution is very likely to be embarrassingly modest, it could offer some information regarding the dimensions of this feature.

- Is there any information behind the hypothesis that the granular material is actually RNA? Some EELS data of HPF-FS sections or the use of a dsRNA antibody could answer this question.

We would like to thank the reviewers for their thorough scrutiny of our manuscript, which we feel has improved it in many ways. We feel confident that we have been able to address the main concerns of the reviewers in a way that should remove their remaining doubts.

Reviewer #1 (Remarks to the Author):

In this manuscript, Dahmane et al. provide the first analysis of enterovirus replication structures by in situ cryo-electron tomography. The authors convincingly show that these virus-induced structures are also involved in enterovirus assembly, and that there is some selective packaging of RNA-loaded virions into the virus-induced double-membrane vesicles (DMVs, in this study termed autophagy-like membranes, ALMs). Moreover, the authors also investigate the role of the autophagy players and establish that, while canonical autophagy may not be crucial, the autophagy factor VPS34 facilitates virus encapsidation, whereas ULK1 may counteract virus assembly and release. The data analysis is sound, the images are breathtaking and show an exquisite level of detail. Although many questions remain open at a mechanistic level and even the identity of some newly-visualized densities is unclear, in my view, this is a wonderful example of how this novel methodology can provide new insight on viral infection, naturally opening a variety of follow-up questions.

Thanks very much for this appreciative overall assessment of our work.

Specific comments:

• I find the use of the term autophagy often rather confusing, starting with the title. It is fine to call the virus-induced double-membrane vesicles ALMs due to their morphological resemblance with autophagosomes, but these appear to be quite different functionally (not for degradation but, at least in part, for the release of virus particles). Moreover, as stressed in this study, it does not seem that canonical autophagy plays a specific role in the formation of ALMs. It would be good to find alternative formulations that make this distinction clear. For example, in the title, “selective autophagy of enteroviruses” is very unclear (“selective engulfment in ALMs”?). Other examples of confusing terms: “virus-induced autophagy”, “autophagic membranes l. 182”, “phagophores” l. 187, 254-5, “autophagic engulfment” l. 274). It would be good to thoroughly revise the text with this in mind.

Thanks for mentioning this. As for DMV-inducing positive-sense RNA viruses, some viruses such as coronaviruses are certainly inducing DMVs that are distinct from autophagosomes. However, the dependence of enterovirus-induced DMVs on the autophagy machinery is strongly established in the literature (e.g. studies by the Kirkegaard group such as PMID: 30608919). The release of enteroviruses in vesicles is also firmly established to take place through the secretory autophagy pathway (reviewed in PMID: 29558400). We also show in this study (Figure S4) that DMV formation is reduced in LC3 KO cells.

We thus agree that this is not canonical autophagy (since it is not induced by the protein kinase ULK1), but as more and more types of selective and non-canonical autophagy have been discovered a picture is emerging where most cellular autophagy may in fact also not be ULK1-induced. We see solid evidence from our study and previous studies of the autophagic identity of enterovirus-induced DMVs. We thus feel that it would be an unwarranted break with existing literature to remove the term autophagy completely. But we do agree that our presentation could be improved. We have thus changed the manuscript title to say “selective secretory autophagy” to underscore that this is not degradative autophagy. We also changed the text on

lines 58, 64 of the introduction to be more precise in this regard, as well as replacing several instances of “autophagic membranes” with “autophagy-like membranes” (lines 198, 202, 228, 235, 858) as suggested by the reviewer.

• Although some EM studies on enterovirus-remodelled membranes are cited, the study on poliovirus (Belov et al. PMID 22072780) is inexplicably missing; please, add. This study (and reference 11) provided data supporting that ALMs arise from the single membrane structures. In the article, this is not mentioned and both types of virus-induced structures appear as completely independent. Importantly, this may have implications for the selective engulfment of RNA-loaded viral capsids into DMVs/ALMs. This could well occur concomitantly with the single to double membrane transformation, instead of the totally independent recruitment that the text and the model in Fig. 6 suggest, which, indeed presents some conundrums (l. 255).

We thank the reviewer for pointing out the omitted reference which we have now added. We have also amended the text to mention the possibility that DMVs arise from single membranes, on lines 60-62.

• Enterovirus viral RNA synthesis is associated with both single-membrane structures and ALMs (Belov et al., Melia et al. PMID 29045829) and not only with single membrane structures as l. 49 suggests. This is perhaps not surprising considering that one type seems to arise from the other, as mentioned in the previous point. Nicely, the current data show that encapsidation, likely linked to replication (ref. 28), can occur on both types of membrane structures.

Thanks for pointing this out. We changed the text to make more clear that RNA replication is known to be associated with both single and double membranes (line 49-50).

• VPS34 is a subunit of the class III PI3K complex formed in the initial steps of autophagy, but plays other roles, for example, in the endocytic pathway. Inhibiting other subunits of the class III PI3K complex would be a great addition, but considering the time-consuming cryo-tomography workflow, this would be understandably outside the scope of this work. However, these are points worth including in the discussion. Along the same lines, there is no direct evidence in this study that virus-induced studies are PI3P-decorated (l. 149), so it would be appropriate to tone down this statement.

This is a good point. We introduced a mention of the other subunits of the PI3K complex in the discussion, lines 264-266. As for the statement about PI3P-decoration, we agree and replaced “PI(3)P-decorated remodeled membranes” with “VPS34 activity” (line 160).

• L. 79. “Requires” appears too strong. The formulation of this point later in the text (l. 109) appears more precise. Even though the data is highly suggestive, as far as I can tell, it cannot be formally excluded that alternative, non-membrane associated, encapsidation paths co-exist. Although this is not explicitly mentioned, I assume that regions containing virus-induced membrane structures were mainly (if not only) selected for data collection, and therefore, not only the sampling is not random, but other regions of the cytoplasm have not been analysed or not in so much detail.

We agree with this. We changed the paragraph title to instead use the expression “correlates with” (line 80). As for the regions selected, they were mainly based on good local lamella quality in the cytoplasm, since membranes were abundant almost anywhere.

• *Old studies (e.g. Belov et al & ref. 11), also reported the presence of multilamellar structures, which appeared to be further enwrapped ALMs. Were such structures completely absent in the samples? I think I see a glimpse of one in Figure S1.*

Indeed, we sometimes did. Sparsely in untreated infected cells, but (by impression) more commonly in ULK1-inhibited infected cells. We added a mention of this in the text (lines 183-185) along with a reference to the published study.

• *The legend of supplementary figure S3 appears to be missing in the uploaded submission.*

Thanks. We added it back.

Reviewer #2 (Remarks to the Author):

In the present manuscript, Dahmane and co-workers describe features of poliovirus assembly in infected cells by using a cryo-electron tomography-based approach. The authors provide compelling evidence that poliovirus capsids assemble predominantly on the cytoplasmic surface of SMs and ALMs, assumed to carry the viral replication machinery. Among the findings, the authors identify a prominent half-capsid assembly intermediate, progression from which to complete virus particles depends on the class III phosphatidylinositol 3-kinase (Vps34), but not the canonical autophagy inducer ULK1/2. Interestingly, inhibiting ULK1/2 strongly increases the appearance of arrays of assembled poliovirus particles. On the cryo-EM part, the authors acquired ~200 tomograms and complemented the qualitative analysis with subtomogram averaging of viral capsids and filaments in the context of virus-induced vesicles.

Overall, the study addresses an interesting and novel aspect of virology, with interest for a broad readership. To the most part, the study has been performed well and with great care. However, several parts require important additional experimentation to ascertain the main findings of the study.

The major concerns are:

1. A major limitation of this study is the lack of detection of viral RNA and the viral replicase machinery. It is claimed that newly formed viral RNA is packaged into capsids. However, there is no data on detection of viral RNA (e.g. by using FISH-based approach) and the viral replicase (e.g. by super-resolution microscopy or immunolabelling of the RdRp or other component of the replicase). One would assume that the replicase sits close to the site where viral RNA is transferred to the half capsids. Alternatively, viral RNA is amplified somewhere else on the membrane surface and then transported to the site of capsid formation. From my point of view, this should be addressed to support the authors claims and the model.

We agree with the reviewer that the structural coordination between capsid assembly and replication complex is the big open question in this study. We would certainly like to be able to answer this, but technical limitations make it impossible at this point. We would like to explain why we have arrived at that conclusion: Firstly, as the reviewer correctly seems to assume the cryo-tomograms don't have the resolution necessary to ascertain the identity of single small proteins such as the RdRP 3D, and RNA is not seen in defined conformations in the tomograms (only as "gray level" inside capsids). This means non-cryo methods that allow for more labelling options would be necessary. However, even our optimised freeze-substitution protocol (Fig S5C-D) was only good enough to visualise the extended virus arrays. It did not allow visualisation of less robust features such as capsid intermediates nor the autophagosome contents (e.g. protein filament bundles). Hence, cryo-ET is until now the only method by which we can specifically identify the half-capsid assembly intermediates in cells, but it does not (currently) allow for identification of nascent viral RNA or the RdRP.

That the single and double membrane vesicles are sites of viral RNA replication is established in previous studies (which on the other hand did not allow for resolution of capsid intermediates). We extended the introduction to mention this (lines 49-50) and added to the discussion a mention of the outstanding question of whether replication and capsid assembly might be structurally linked (lines 254-256). We hope that the reviewer will appreciate that we find this question important and indeed wish to return to it as soon as we find a viable approach, but simply view it as experimentally intractable at the moment.

2. Figure 4 A-G: There is no information about the tethering of putative PV particles in the context of ULK-inhibition. In the images provided, it seems that no tethered particles are found, but given the

increase in infectivity, the assembly of the particles should also increase. Is the absence (lack of detection) of tethered particles due to very much accelerated assembly?

Good point. In response to this question we did the analysis and added the outcome to Fig. S5B (discussed in the text on lines 178-179). As expected this shows a decrease in the fraction of tethered particles, mainly being due to the fact that the particles in arrays are no longer membrane-proximal nor tethered. Since we still do see tethered particles we have no reason to believe that the assembly mechanism has changed, so it might be a question of kinetics as suggested by the reviewer.

3. Line 116: the authors mention variable levels of densities in the intermediate capsid structures (Fig 2B-E). At the shown resolution, it is difficult to discern if these densities are different or are RNA. The authors should validate in an alternative way whether or not the lumen of assembled capsids contains viral RNA. This is also the case for Figure 4I: The authors could prove via immunolabelling or negative staining that the electron-dense round structures are indeed PVs inside the extracellular vesicles or viral RNA inside the newly encapsidated structures.

As for the capsid intermediates, we unfortunately cannot know (see above). The contents might well be replicase proteins at that point. In general, as for the empty and full capsids: we believe the hydantoin treatment serves as an “internal control” that what we identify as empty capsids are indeed RNA-less capsids. Hydantoin is well characterised to act as an encapsidation inhibitor, and the empty particles we see in those tomograms are indistinguishable from empty capsids seen in untreated tomograms.

As for Figure 4I: in response to the reviewer’s request we collected additional tomography data on ULK-inhibited cells. This time we did not do FIB-milling, but instead focussed on the peripheral plasma membrane of intact cells. Interestingly, we found examples of released RNA-loaded virions in vesicles, just outside the plasma membrane. We have added this to the updated Fig. 4J-N and mention it in the text on lines 192-194. This observation correlates well with a tomography study on purified extracellular vesicles from PV-infected cells (Yang et al, Sci Rep 2020, PMID: 32409751) referenced by us. Note that it is also established by previous studies that vesicle release from infected cells in non-lytic (e.g. papers by Altan-Bonnet group).

4. Figure 1, Figure 5: The authors claim assembly occurs on autophagy membranes but there is no marker for this. Perhaps it would be possible to use the same Kukulski protocol used for the thin sections analysis to perform CLEM with a fluorescent marker for autophagy? In addition, the authors could prove via immunolabelling or negative staining that the electron-dense round structures are indeed PVs inside the extracellular vesicles.

This is a good point. The Kukulski protocol unfortunately could not visualise capsid intermediates, only virus arrays, as explained above. Instead, to address the reviewer’s request we performed co-IF with an anti-LC3 antibody (as autophagy membrane marker) and the conformation-specific monoclonal antibody A12 which only recognises assembled capsid protein, interacting with the canyon region of the capsid and capsomers. These new data, shown in the updated Fig S3 F-G, show a clear colocalisation of the signals for LC3 and assembled capsid (/intermediate). We mention the data in the text on lines 122-125.

5. Figure 3F: The author should include MRT68921 treatment only and also measure the specific infectivity of extracellular viruses (e.g measure the amount of extracellular viral capsid, viral RNA) to clarify the decrease in virus release or virus infectivity in VPS34-treated cells.

The requested data on MRT68921 treatment only was actually already in Fig. 4H, we thought it was thematically better to place it together with the other ULK-related data. Moreover, in response to the reviewer's question we performed qPCR assays on extracellular fractions from PV-infected cells with and without drug treatment to address the question whether virus release of virus infectivity is decreased (Figure S4K). The amount of extracellular viral RNA is quasi-absent in the presence of VPS34 inhibitor as compared to DMSO and MRT-treated cells, highlighting that there is a decrease in virus release, not infectivity of released virions.

6. Line 138: The authors claim that VPS34 inhibition does not affect viral RNA replication. However, Figure S4G shows some reduction in intracellular viral RNA in VSP34 treated cells although it is not statistically significant. In addition, by admission of the authors VPS34 is required for high level membrane proliferation. A proper discussion of this observation is required.

In response to request 7 below we conducted more extensive measurements of intracellular RNA at different MOIs and different drug treatments. In fact, these data led to a reinterpretation of the effect of VPS34. We now see (as the reviewer guessed from the previous data) that VPS34 has an overall inhibitory effect on viral RNA replication (only significant at the lowest MOI). This slightly changes our interpretation of the role of VPS34, but not of the importance of the half-capsid assembly intermediate and the fact that it is membrane localised. The changes to the text as a result of these new data are found on lines 147-150, 258-259.

7. Into the same vein, please include MRT68921 treatment only and also a positive control (e.g. guanidine-HCl). In this experiment, different MOIs should also be tested (e.g. 1, 0.1...).

We performed the requested experiments. The data are now in the new figure S4H. They confirm that hydantoin does not affect viral RNA replication, and that GnHCl inhibits it (positive control). The only significant differences in intracellular RNA for any drug treatment is an increase in intracellular viral RNA upon ULK inhibition and a reduction upon Vps34-IN1 treatment (both only significant at the lowest MOI 0.1). Based on the result for VPS34 we decided to revise certain parts of the text. See response to comment 6 above.

Minor points:

1. The authors state the membrane tethering complex is regular in length. Did the authors consider trying a further characterization, perhaps by subtomogram averaging?

This is an excellent suggestion. We managed to find and average ~200 tethered particles from the tomograms. Although the resolution of the subtomogram average is highly limited by particle number, the average does provide an envelope that allows us to estimate the dimensions and molecular mass of the tether. We feel that this provides an important initial characterisation of the tether and added it to Fig. 1F, with a text mention on lines 96-98.

2. Line 157 typo, inhibition with capital I

Thanks. Fixed.

3. Line 421: Tomogram averaging of empty and full capsids: Is there any structural difference in the capsid, except RNA loading? Did the authors observe the icosahedral structure of PV capsid in any of the assembled structures?

In response to this request and a request from another reviewer we made additional efforts in subtomogram averaging. We picked more particles and improved the subtomogram averaging to include a classification step. The workflow is described in the new figure S9. While the resolution of the capsids increased slightly, it unfortunately did not increase enough to be able to answer the questions raised here with any acceptable certainty. The challenge here is that enterovirus capsids are small (from a cryo-ET perspective) and “smooth” at lower resolutions (no surface spikes etc to drive alignment). This makes it challenging to get the precision in subtomogram alignment which would be needed to reach secondary-structure resolution, at which we could address the questions raised with acceptable certainty. We enforced icosahedral symmetry on the reconstruction but were not able to resolve the level of detail that would allow us to address the interesting question of RNA-induced capsid conformation changes. From X-ray and single-particle structures the structural differences related to RNA loading are expected to be on the inner face of the capsid and would require higher resolution to resolve. Thus, we prefer to be conservative and not discuss the very slight differences in our low-resolution averages of empty and full capsids in the article. We hope the reviewer can appreciate our reasoning in this case. We are keen to return to the question but would need even higher data quality, perhaps dependent on improved detectors or data acquisition schemes.

4. Line 490: Please explain why you used coxsackievirus for template matching rather than the structure calculated by the virion arrays. Is there any difference between the published structure and the one determined by subtomogram averaging?

The reason was as simple technical one: the coxsackie study had both empty and full particle structures determined by EM in the same study and the same microscope settings. They would thus be more comparable (pixel size etc) to use as templates. Prior to use as templates, both references were low-pass filtered beyond the point where they will be indistinguishable from poliovirus capsids.

5. Line 794: wrong colour code for ALM (purple). It should be blue.

Thanks for the comment. However, we have polled a few reference persons who were all of the opinion that this is purple. Also, the structures we designate as “dense granules” are in light blue. We thus feel it more consistent to keep the current colour designation, but would be open to changing it during the typesetting process.

6. Line 838: typo “tomigram”

Thanks for finding this. Fixed.

7. Line 421: Tomogram averaging of empty and full capsids: Is there any structural difference in the capsid, except RNA loading? The icosahedral structure could be shown? Are empty capsids known to have some biological function or are they just a byproduct? Please comment.

As for structural differences, see response to minor point 3 above. We are not aware of any specific roles for empty capsids.

8. Fig 3F: If vesicular release is affected in ULK1/2 inhibited cells, intracellular infectivity should also be higher. Did the authors measure this?

Good point. In response to this request we attempted the measurement (see below) but there was no significant difference in measured intracellular TCID₅₀ at 6 h p.i. Since there are clearly more intracellular virions in the tomograms, and more release of infectious particles at 8 h p.i., we think the most likely explanation is that our TCID₅₀ measurement protocol might pellet the large virus arrays in the pelleting intended to remove large cellular debris (or give a subset of cells a very high MOI). Thus we would prefer not to include this inconclusive measurement in the paper.

9. Figure 5: EM images have very strong contrast and very little grey level. It would be helpful to see the images at a lower binning.

We agree, thanks for pointing this out. We adjusted figure 5.

10. Table S2, line 568: numbers seem to be in contradiction of the data shown in Figure 1 plot E and K. Perhaps rows 1-3 were inverted?

Indeed, thanks for finding this. We corrected the mistake.

11. Figure S4A. The author should include VPS34 treatment only and include other markers of autophagy e.g. p62. The electron-dense dots budded from the SMs and ALMs, and in the vicinity of these membranes (Figure 1), were identified as PV particles. This observation was confirmed by Hydantoin treatment to prevent RNA loading of PV capsids. However, these budding “virions” are rather reminiscent of spherules that are produced next to ALMs by MERS-CoV and infectious bronchitis virus (IBV). This point should be discussed.

We redid the western blots according to the suggestions, to include Vps34-IN1 treatment only (which had a similar level of LC3 lipidation as VPS34 + MRT). We also added p62 which is found to be characteristically cleaved upon polio infection in all treatments. The new blots are in Fig S4A.

We have looked at the paper on the coronavirus IBV which we think the reviewer is referring to (PMID: 24149513). The spherules in that article are clearly membrane invaginations as seen e.g. in their Fig 3E. On the other hand, our half-capsid intermediates are discontinuous with the membrane and appear thinner than the membrane, which rules out that they are lipid membrane. We did not mention this clearly in the manuscript and would like to thank the reviewer for pointing that out. We now describe it on lines 119-122.

12. Table S2, line 568: Numbers seem to be in contradiction of the data shown in Figure 1 plot E and K. Perhaps rows 1-3 were inverted?

Fixed, see comment to minor point 10 above.

Reviewer #3 (Remarks to the Author):

The manuscript by Dahmane and colleagues attacks, from a structural perspective, one of the most difficult questions in virology – how are nonenveloped RNA virions assembled in the cytoplasm of infected cells. While single particle cryo-electron microscopy has opened the gates for a deluge of structures describing various intermediates associated with viral replication, understanding the behaviour of these proteins in their cellular context is still a difficult task. The present study uses the laborious strategy of preparing FIB cryo lamellae from infected cells and employs cryo-electron tomography to reconstruct the three-dimensional landscape of specific cellular compartments. The model of choice is poliovirus – the most studied member of enteroviruses – for which a large volume of biochemical information is available regarding the packaging of mature virions and their release either through the common lytic process or through the shedding of virions wrapped in cytoplasmic membranes.

The present article is a tour de force from a microscopic point of view and clarifies a number of long standing questions in the picornavirus field. First, the authors show that poliovirus capsids are packaged if and only if they are attached to cytoplasmic membranes. Secondly, they identify, in-situ, two important assembly intermediates: the first, corresponds to roughly a half capsid and is associated with genome packaging, while the second is tethered at a constant distance from the membranes. Both of these structures are present only on the cytosolic side of the membrane and are absent in the luminal space. Thirdly, they show that there is a strong selectivity for full virions in cytoplasmic vesicles. Finally, the authors find cellular factors involved in the progression over the half capsid intermediate, factors regulating the release of cargos transporting virions via the autophagy pathway, and factors controlling the formation of intracellular viral arrays and virion release. The results are supported by solid statistics obtained from a reasonable number of tomograms obtained from cellular lamellae obtained under different conditions.

Overall, I consider that this manuscript will be of extraordinary importance for virologists and it will open a rich avenue for further studies. I warmly recommend its publication.

I have few minor points that, if addressed by authors, would improve the manuscript.

We would like to thank the reviewer for this very positive assessment of our work.

- The authors have performed subtomogram averaging of full and empty capsids imposing icosahedral symmetry. However, despite the larger number of full capsids the resolution attained was lower than for empty capsids. Did the authors try to see if more than one assembly intermediate can be identified in the population of full capsids? Do all capsids show the same amount of internal density? I suggest the authors to include a supplemental figure with the capsid analysis.

This is a good suggestion. We extended the STA of the virions with more particles (still limited to ~500 each) and the processing is now described in the new supplementary figure S9. In the new processing scheme we included a classification round. However, classification was basically just able to weed out “bad particles”, not credible alternative conformations as the reviewer was hoping for. For 500 icosahedral particles the attained resolution might seem underwhelming. We think this is due to the simple fact that the poliovirus particles are so small (at 30 nm diameter they are hardly larger than a ribosome, and smaller than virus capsids previously averaged in situ by STA) and so smooth (no spikes or other low-res features to help drive alignment). We thus believe that the current tomograms don’t have enough signal to drive the alignment of isolated capsids to the point that we can start to see secondary structure.

- The presence of the membrane associated half-capsid intermediate is a very intriguing finding of this study. While I realize that there is a small number of subtomogram volumes trapping this assembly stage, I wonder if the authors have tried to see if there is any symmetry related information in these

particles. Is a 5-fold axis perpendicular to the membrane? Using the icosahedral reconstruction would be possible to obtain a missing wedge corrected cross correlation value for each of these volumes.
- The same question regarding the tethered capsids.

This is an important question and one that we tried to answer in response to this comment. We indeed believe that capsid intermediates and tethered capsids interact with membranes and tethers at specific sites. But we ran into a version of the problems outline above: the capsids don't seem to contain enough shape signal (i.e. too small and too smooth) to do this measurement. As an example of what we tried, below is the outcome when we aligned all individual tethered particles to a single-particle enterovirus structure. The red dots representing the tether position appear nearly randomly distributed which we attribute to insufficient signal.

Response Figure 2: Identified tether position (red dots) by cross-correlation of individual tethered particle subtomograms with a poliovirus reference (grey).

- Did the authors attempt the reconstruction of the tethering domain? While the resolution is very likely to be embarrassingly modest, it could offer some information regarding the dimensions of this feature.

We are not embarrassed by low resolution if the biological insights are good, and thus took the reviewer up on this excellent suggestion. We managed to pick and average ~200 tethered particles from the tomograms. Although the resolution of the average is indeed limited by particle number, the average provides an envelope that allows us to estimate the dimensions and molecular mass of the tether. We feel that this provides an important initial characterisation of the tether and added it to the new Fig. 1F, and lines 96-98.

- Is there any information behind the hypothesis that the granular material is actually RNA? Some EELS data of HPF-FS sections or the use of a dsRNA antibody could answer this question.

We tried to pursue this question by HPF-FS and immunolabelling. However, even with protocols optimised to retain the morphology of other autophagic structures (p62 assemblies) we could not distinguish different types of autophagosome contents (neither single virions, granules, nor the filament bundles). Thus, attempts at immunolabelling would make no sense. We have altered the text on lines 283-284 to reflect the uncertainty of the identity of the granules.

Reviewer Comments, second round

Reviewer #1 (Remarks to the Author):

The authors have sufficiently addressed the points that I raised for the first version of the manuscript.

Reviewer #2 (Remarks to the Author):

In this revised manuscript by Dahmane and colleagues, the authors have carefully addressed all comments made by this reviewer. Additional experiments are fully convincing as are the given explanations why some of the raised questions cannot be addressed at this current stage. Overall, I consider this an excellent and highly informative study for which the authors should be applauded. I am convinced that many people will read this paper with great interest.

Reviewer #3 (Remarks to the Author):

The authors have put a lot of effort into improving the manuscript. I consider that in the present form the study is ready to be published.

I want to congratulate the authors for this beautiful work.

Mihnea Bostina

Otago, New Zealand

We would like to thank the reviewers for their thorough scrutiny of our manuscript, which we feel has improved it in many ways. We feel confident that we have been able to address the main concerns of the reviewers in a way that should remove their remaining doubts.

Reviewer #1 (Remarks to the Author):

In this manuscript, Dahmane et al. provide the first analysis of enterovirus replication structures by in situ cryo-electron tomography. The authors convincingly show that these virus-induced structures are also involved in enterovirus assembly, and that there is some selective packaging of RNA-loaded virions into the virus-induced double-membrane vesicles (DMVs, in this study termed autophagy-like membranes, ALMs). Moreover, the authors also investigate the role of the autophagy players and establish that, while canonical autophagy may not be crucial, the autophagy factor VPS34 facilitates virus encapsidation, whereas ULK1 may counteract virus assembly and release. The data analysis is sound, the images are breathtaking and show an exquisite level of detail. Although many questions remain open at a mechanistic level and even the identity of some newly-visualized densities is unclear, in my view, this is a wonderful example of how this novel methodology can provide new insight on viral infection, naturally opening a variety of follow-up questions.

Thanks very much for this appreciative overall assessment of our work.

Specific comments:

• I find the use of the term autophagy often rather confusing, starting with the title. It is fine to call the virus-induced double-membrane vesicles ALMs due to their morphological resemblance with autophagosomes, but these appear to be quite different functionally (not for degradation but, at least in part, for the release of virus particles). Moreover, as stressed in this study, it does not seem that canonical autophagy plays a specific role in the formation of ALMs. It would be good to find alternative formulations that make this distinction clear. For example, in the title, “selective autophagy of enteroviruses” is very unclear (“selective engulfment in ALMs”?). Other examples of confusing terms: “virus-induced autophagy”, “autophagic membranes l. 182”, “phagophores” l. 187, 254-5, “autophagic engulfment” l. 274). It would be good to thoroughly revise the text with this in mind.

Thanks for mentioning this. As for DMV-inducing positive-sense RNA viruses, some viruses such as coronaviruses are certainly inducing DMVs that are distinct from autophagosomes. However, the dependence of enterovirus-induced DMVs on the autophagy machinery is strongly established in the literature (e.g. studies by the Kirkegaard group such as PMID: 30608919). The release of enteroviruses in vesicles is also firmly established to take place through the secretory autophagy pathway (reviewed in PMID: 29558400). We also show in this study (Figure S4) that DMV formation is reduced in LC3 KO cells.

We thus agree that this is not canonical autophagy (since it is not induced by the protein kinase ULK1), but as more and more types of selective and non-canonical autophagy have been discovered a picture is emerging where most cellular autophagy may in fact also not be ULK1-induced. We see solid evidence from our study and previous studies of the autophagic identity of enterovirus-induced DMVs. We thus feel that it would be an unwarranted break with existing literature to remove the term autophagy completely. But we do agree that our presentation could be improved. We have thus changed the manuscript title to say “selective secretory autophagy” to underscore that this is not degradative autophagy. We also changed the text on

lines 58, 64 of the introduction to be more precise in this regard, as well as replacing several instances of “autophagic membranes” with “autophagy-like membranes” (lines 198, 202, 228, 235, 858) as suggested by the reviewer.

• Although some EM studies on enterovirus-remodelled membranes are cited, the study on poliovirus (Belov et al. PMID 22072780) is inexplicably missing; please, add. This study (and reference 11) provided data supporting that ALMs arise from the single membrane structures. In the article, this is not mentioned and both types of virus-induced structures appear as completely independent. Importantly, this may have implications for the selective engulfment of RNA-loaded viral capsids into DMVs/ALMs. This could well occur concomitantly with the single to double membrane transformation, instead of the totally independent recruitment that the text and the model in Fig. 6 suggest, which, indeed presents some conundrums (l. 255).

We thank the reviewer for pointing out the omitted reference which we have now added. We have also amended the text to mention the possibility that DMVs arise from single membranes, on lines 60-62.

• Enterovirus viral RNA synthesis is associated with both single-membrane structures and ALMs (Belov et al., Melia et al. PMID 29045829) and not only with single membrane structures as l. 49 suggests. This is perhaps not surprising considering that one type seems to arise from the other, as mentioned in the previous point. Nicely, the current data show that encapsidation, likely linked to replication (ref. 28), can occur on both types of membrane structures.

Thanks for pointing this out. We changed the text to make more clear that RNA replication is known to be associated with both single and double membranes (line 49-50).

• VPS34 is a subunit of the class III PI3K complex formed in the initial steps of autophagy, but plays other roles, for example, in the endocytic pathway. Inhibiting other subunits of the class III PI3K complex would be a great addition, but considering the time-consuming cryo-tomography workflow, this would be understandably outside the scope of this work. However, these are points worth including in the discussion. Along the same lines, there is no direct evidence in this study that virus-induced studies are PI3P-decorated (l. 149), so it would be appropriate to tone down this statement.

This is a good point. We introduced a mention of the other subunits of the PI3K complex in the discussion, lines 264-266. As for the statement about PI3P-decoration, we agree and replaced “PI(3)P-decorated remodeled membranes” with “VPS34 activity” (line 160).

• L. 79. “Requires” appears too strong. The formulation of this point later in the text (l. 109) appears more precise. Even though the data is highly suggestive, as far as I can tell, it cannot be formally excluded that alternative, non-membrane associated, encapsidation paths co-exist. Although this is not explicitly mentioned, I assume that regions containing virus-induced membrane structures were mainly (if not only) selected for data collection, and therefore, not only the sampling is not random, but other regions of the cytoplasm have not been analysed or not in so much detail.

We agree with this. We changed the paragraph title to instead use the expression “correlates with” (line 80). As for the regions selected, they were mainly based on good local lamella quality in the cytoplasm, since membranes were abundant almost anywhere.

• *Old studies (e.g. Belov et al & ref. 11), also reported the presence of multilamellar structures, which appeared to be further enwrapped ALMs. Were such structures completely absent in the samples? I think I see a glimpse of one in Figure S1.*

Indeed, we sometimes did. Sparsely in untreated infected cells, but (by impression) more commonly in ULK1-inhibited infected cells. We added a mention of this in the text (lines 183-185) along with a reference to the published study.

• *The legend of supplementary figure S3 appears to be missing in the uploaded submission.*

Thanks. We added it back.

Reviewer #2 (Remarks to the Author):

In the present manuscript, Dahmane and co-workers describe features of poliovirus assembly in infected cells by using a cryo-electron tomography-based approach. The authors provide compelling evidence that poliovirus capsids assemble predominantly on the cytoplasmic surface of SMs and ALMs, assumed to carry the viral replication machinery. Among the findings, the authors identify a prominent half-capsid assembly intermediate, progression from which to complete virus particles depends on the class III phosphatidylinositol 3-kinase (Vps34), but not the canonical autophagy inducer ULK1/2. Interestingly, inhibiting ULK1/2 strongly increases the appearance of arrays of assembled poliovirus particles. On the cryo-EM part, the authors acquired ~200 tomograms and complemented the qualitative analysis with subtomogram averaging of viral capsids and filaments in the context of virus-induced vesicles.

Overall, the study addresses an interesting and novel aspect of virology, with interest for a broad readership. To the most part, the study has been performed well and with great care. However, several parts require important additional experimentation to ascertain the main findings of the study.

The major concerns are:

1. A major limitation of this study is the lack of detection of viral RNA and the viral replicase machinery. It is claimed that newly formed viral RNA is packaged into capsids. However, there is no data on detection of viral RNA (e.g. by using FISH-based approach) and the viral replicase (e.g. by super-resolution microscopy or immunolabelling of the RdRp or other component of the replicase). One would assume that the replicase sits close to the site where viral RNA is transferred to the half capsids. Alternatively, viral RNA is amplified somewhere else on the membrane surface and then transported to the site of capsid formation. From my point of view, this should be addressed to support the authors claims and the model.

We agree with the reviewer that the structural coordination between capsid assembly and replication complex is the big open question in this study. We would certainly like to be able to answer this, but technical limitations make it impossible at this point. We would like to explain why we have arrived at that conclusion: Firstly, as the reviewer correctly seems to assume the cryo-tomograms don't have the resolution necessary to ascertain the identity of single small proteins such as the RdRP 3D, and RNA is not seen in defined conformations in the tomograms (only as "gray level" inside capsids). This means non-cryo methods that allow for more labelling options would be necessary. However, even our optimised freeze-substitution protocol (Fig S5C-D) was only good enough to visualise the extended virus arrays. It did not allow visualisation of less robust features such as capsid intermediates nor the autophagosome contents (e.g. protein filament bundles). Hence, cryo-ET is until now the only method by which we can specifically identify the half-capsid assembly intermediates in cells, but it does not (currently) allow for identification of nascent viral RNA or the RdRP.

That the single and double membrane vesicles are sites of viral RNA replication is established in previous studies (which on the other hand did not allow for resolution of capsid intermediates). We extended the introduction to mention this (lines 49-50) and added to the discussion a mention of the outstanding question of whether replication and capsid assembly might be structurally linked (lines 254-256). We hope that the reviewer will appreciate that we find this question important and indeed wish to return to it as soon as we find a viable approach, but simply view it as experimentally intractable at the moment.

2. Figure 4 A-G: There is no information about the tethering of putative PV particles in the context of ULK-inhibition. In the images provided, it seems that no tethered particles are found, but given the

increase in infectivity, the assembly of the particles should also increase. Is the absence (lack of detection) of tethered particles due to very much accelerated assembly?

Good point. In response to this question we did the analysis and added the outcome to Fig. S5B (discussed in the text on lines 178-179). As expected this shows a decrease in the fraction of tethered particles, mainly being due to the fact that the particles in arrays are no longer membrane-proximal nor tethered. Since we still do see tethered particles we have no reason to believe that the assembly mechanism has changed, so it might be a question of kinetics as suggested by the reviewer.

3. Line 116: the authors mention variable levels of densities in the intermediate capsid structures (Fig 2B-E). At the shown resolution, it is difficult to discern if these densities are different or are RNA. The authors should validate in an alternative way whether or not the lumen of assembled capsids contains viral RNA. This is also the case for Figure 4I: The authors could prove via immunolabelling or negative staining that the electron-dense round structures are indeed PVs inside the extracellular vesicles or viral RNA inside the newly encapsidated structures.

As for the capsid intermediates, we unfortunately cannot know (see above). The contents might well be replicase proteins at that point. In general, as for the empty and full capsids: we believe the hydantoin treatment serves as an “internal control” that what we identify as empty capsids are indeed RNA-less capsids. Hydantoin is well characterised to act as an encapsidation inhibitor, and the empty particles we see in those tomograms are indistinguishable from empty capsids seen in untreated tomograms.

As for Figure 4I: in response to the reviewer’s request we collected additional tomography data on ULK-inhibited cells. This time we did not do FIB-milling, but instead focussed on the peripheral plasma membrane of intact cells. Interestingly, we found examples of released RNA-loaded virions in vesicles, just outside the plasma membrane. We have added this to the updated Fig. 4J-N and mention it in the text on lines 192-194. This observation correlates well with a tomography study on purified extracellular vesicles from PV-infected cells (Yang et al, Sci Rep 2020, PMID: 32409751) referenced by us. Note that it is also established by previous studies that vesicle release from infected cells in non-lytic (e.g. papers by Altan-Bonnet group).

4. Figure 1, Figure 5: The authors claim assembly occurs on autophagy membranes but there is no marker for this. Perhaps it would be possible to use the same Kukulski protocol used for the thin sections analysis to perform CLEM with a fluorescent marker for autophagy? In addition, the authors could prove via immunolabelling or negative staining that the electron-dense round structures are indeed PVs inside the extracellular vesicles.

This is a good point. The Kukulski protocol unfortunately could not visualise capsid intermediates, only virus arrays, as explained above. Instead, to address the reviewer’s request we performed co-IF with an anti-LC3 antibody (as autophagy membrane marker) and the conformation-specific monoclonal antibody A12 which only recognises assembled capsid protein, interacting with the canyon region of the capsid and capsomers. These new data, shown in the updated Fig S3 F-G, show a clear colocalisation of the signals for LC3 and assembled capsid (intermediate). We mention the data in the text on lines 122-125.

5. Figure 3F: The author should include MRT68921 treatment only and also measure the specific infectivity of extracellular viruses (e.g measure the amount of extracellular viral capsid, viral RNA) to clarify the decrease in virus release or virus infectivity in VPS34-treated cells.

The requested data on MRT68921 treatment only was actually already in Fig. 4H, we thought it was thematically better to place it together with the other ULK-related data. Moreover, in response to the reviewer's question we performed qPCR assays on extracellular fractions from PV-infected cells with and without drug treatment to address the question whether virus release of virus infectivity is decreased (Figure S4K). The amount of extracellular viral RNA is quasi-absent in the presence of VPS34 inhibitor as compared to DMSO and MRT-treated cells, highlighting that there is a decrease in virus release, not infectivity of released virions.

6. Line 138: The authors claim that VPS34 inhibition does not affect viral RNA replication. However, Figure S4G shows some reduction in intracellular viral RNA in VSP34 treated cells although it is not statistically significant. In addition, by admission of the authors VPS34 is required for high level membrane proliferation. A proper discussion of this observation is required.

In response to request 7 below we conducted more extensive measurements of intracellular RNA at different MOIs and different drug treatments. In fact, these data led to a reinterpretation of the effect of VPS34. We now see (as the reviewer guessed from the previous data) that VPS34 has an overall inhibitory effect on viral RNA replication (only significant at the lowest MOI). This slightly changes our interpretation of the role of VPS34, but not of the importance of the half-capsid assembly intermediate and the fact that it is membrane localised. The changes to the text as a result of these new data are found on lines 147-150, 258-259.

7. Into the same vein, please include MRT68921 treatment only and also a positive control (e.g. guanidine-HCl). In this experiment, different MOIs should also be tested (e.g. 1, 0.1...).

We performed the requested experiments. The data are now in the new figure S4H. They confirm that hydantoin does not affect viral RNA replication, and that GnHCl inhibits it (positive control). The only significant differences in intracellular RNA for any drug treatment is an increase in intracellular viral RNA upon ULK inhibition and a reduction upon Vps34-IN1 treatment (both only significant at the lowest MOI 0.1). Based on the result for VPS34 we decided to revise certain parts of the text. See response to comment 6 above.

Minor points:

1. The authors state the membrane tethering complex is regular in length. Did the authors consider trying a further characterization, perhaps by subtomogram averaging?

This is an excellent suggestion. We managed to find and average ~200 tethered particles from the tomograms. Although the resolution of the subtomogram average is highly limited by particle number, the average does provide an envelope that allows us to estimate the dimensions and molecular mass of the tether. We feel that this provides an important initial characterisation of the tether and added it to Fig. 1F, with a text mention on lines 96-98.

2. Line 157 typo, inhibition with capital I

Thanks. Fixed.

3. Line 421: Tomogram averaging of empty and full capsids: Is there any structural difference in the capsid, except RNA loading? Did the authors observe the icosahedral structure of PV capsid in any of the assembled structures?

In response to this request and a request from another reviewer we made additional efforts in subtomogram averaging. We picked more particles and improved the subtomogram averaging to include a classification step. The workflow is described in the new figure S9. While the resolution of the capsids increased slightly, it unfortunately did not increase enough to be able to answer the questions raised here with any acceptable certainty. The challenge here is that enterovirus capsids are small (from a cryo-ET perspective) and “smooth” at lower resolutions (no surface spikes etc to drive alignment). This makes it challenging to get the precision in subtomogram alignment which would be needed to reach secondary-structure resolution, at which we could address the questions raised with acceptable certainty. We enforced icosahedral symmetry on the reconstruction but were not able to resolve the level of detail that would allow us to address the interesting question of RNA-induced capsid conformation changes. From X-ray and single-particle structures the structural differences related to RNA loading are expected to be on the inner face of the capsid and would require higher resolution to resolve. Thus, we prefer to be conservative and not discuss the very slight differences in our low-resolution averages of empty and full capsids in the article. We hope the reviewer can appreciate our reasoning in this case. We are keen to return to the question but would need even higher data quality, perhaps dependent on improved detectors or data acquisition schemes.

4. Line 490: Please explain why you used coxsackievirus for template matching rather than the structure calculated by the virion arrays. Is there any difference between the published structure and the one determined by subtomogram averaging?

The reason was as simple technical one: the coxsackie study had both empty and full particle structures determined by EM in the same study and the same microscope settings. They would thus be more comparable (pixel size etc) to use as templates. Prior to use as templates, both references were low-pass filtered beyond the point where they will be indistinguishable from poliovirus capsids.

5. Line 794: wrong colour code for ALM (purple). It should be blue.

Thanks for the comment. However, we have polled a few reference persons who were all of the opinion that this is purple. Also, the structures we designate as “dense granules” are in light blue. We thus feel it more consistent to keep the current colour designation, but would be open to changing it during the typesetting process.

6. Line 838: typo “tomigram”

Thanks for finding this. Fixed.

7. Line 421: Tomogram averaging of empty and full capsids: Is there any structural difference in the capsid, except RNA loading? The icosahedral structure could be shown? Are empty capsids known to have some biological function or are they just a byproduct? Please comment.

As for structural differences, see response to minor point 3 above. We are not aware of any specific roles for empty capsids.

8. Fig 3F: If vesicular release is affected in ULK1/2 inhibited cells, intracellular infectivity should also be higher. Did the authors measure this?

Good point. In response to this request we attempted the measurement (see below) but there was no significant difference in measured intracellular TCID₅₀ at 6 h p.i. Since there are clearly more intracellular virions in the tomograms, and more release of infectious particles at 8 h p.i., we think the most likely explanation is that our TCID₅₀ measurement protocol might pellet the large virus arrays in the pelletation intended to remove large cellular debris (or give a subset of cells a very high MOI). Thus we would prefer not to include this inconclusive measurement in the paper.

9. *Figure 5: EM images have very strong contrast and very little grey level. It would be helpful to see the images at a lower binning.*

We agree, thanks for pointing this out. We adjusted figure 5.

10. *Table S2, line 568: numbers seem to be in contradiction of the data shown in Figure 1 plot E and K. Perhaps rows 1-3 were inverted?*

Indeed, thanks for finding this. We corrected the mistake.

11. *Figure S4A. The author should include VPS34 treatment only and include other markers of autophagy e.g. p62. The electron-dense dots budded from the SMs and ALMs, and in the vicinity of these membranes (Figure 1), were identified as PV particles. This observation was confirmed by Hydantoin treatment to prevent RNA loading of PV capsids. However, these budding “virions” are rather reminiscent of spherules that are produced next to ALMs by MERS-CoV and infectious bronchitis virus (IBV). This point should be discussed.*

We redid the western blots according to the suggestions, to include Vps34-IN1 treatment only (which had a similar level of LC3 lipidation as VPS34 + MRT). We also added p62 which is found to be characteristically cleaved upon polio infection in all treatments. The new blots are in Fig S4A.

We have looked at the paper on the coronavirus IBV which we think the reviewer is referring to (PMID: 24149513). The spherules in that article are clearly membrane invaginations as seen e.g. in their Fig 3E. On the other hand, our half-capsid intermediates are discontinuous with the membrane and appear thinner than the membrane, which rules out that they are lipid membrane. We did not mention this clearly in the manuscript and would like to thank the reviewer for pointing that out. We now describe it on lines 119-122.

12. Table S2, line 568: Numbers seem to be in contradiction of the data shown in Figure 1 plot E and K. Perhaps rows 1-3 were inverted?

Fixed, see comment to minor point 10 above.

Reviewer #3 (Remarks to the Author):

The manuscript by Dahmane and colleagues attacks, from a structural perspective, one of the most difficult questions in virology – how are nonenveloped RNA virions assembled in the cytoplasm of infected cells. While single particle cryo-electron microscopy has opened the gates for a deluge of structures describing various intermediates associated with viral replication, understanding the behaviour of these proteins in their cellular context is still a difficult task. The present study uses the laborious strategy of preparing FIB cryo lamellae from infected cells and employs cryo-electron tomography to reconstruct the three-dimensional landscape of specific cellular compartments. The model of choice is poliovirus – the most studied member of enteroviruses – for which a large volume of biochemical information is available regarding the packaging of mature virions and their release either through the common lytic process or through the shedding of virions wrapped in cytoplasmic membranes.

The present article is a tour de force from a microscopic point of view and clarifies a number of long standing questions in the picornavirus field. First, the authors show that poliovirus capsids are packaged if and only if they are attached to cytoplasmic membranes. Secondly, they identify, in-situ, two important assembly intermediates: the first, corresponds to roughly a half capsid and is associated with genome packaging, while the second is tethered at a constant distance from the membranes. Both of these structures are present only on the cytosolic side of the membrane and are absent in the luminal space. Thirdly, they show that there is a strong selectivity for full virions in cytoplasmic vesicles. Finally, the authors find cellular factors involved in the progression over the half capsid intermediate, factors regulating the release of cargos transporting virions via the autophagy pathway, and factors controlling the formation of intracellular viral arrays and virion release. The results are supported by solid statistics obtained from a reasonable number of tomograms obtained from cellular lamellae obtained under different conditions.

Overall, I consider that this manuscript will be of extraordinary importance for virologists and it will open a rich avenue for further studies. I warmly recommend its publication.

I have few minor points that, if addressed by authors, would improve the manuscript.

We would like to thank the reviewer for this very positive assessment of our work.

- The authors have performed subtomogram averaging of full and empty capsids imposing icosahedral symmetry. However, despite the larger number of full capsids the resolution attained was lower than for empty capsids. Did the authors try to see if more than one assembly intermediate can be identified in the population of full capsids? Do all capsids show the same amount of internal density? I suggest the authors to include a supplemental figure with the capsid analysis.

This is a good suggestion. We extended the STA of the virions with more particles (still limited to ~500 each) and the processing is now described in the new supplementary figure S9. In the new processing scheme we included a classification round. However, classification was basically just able to weed out “bad particles”, not credible alternative conformations as the reviewer was hoping for. For 500 icosahedral particles the attained resolution might seem underwhelming. We think this is due to the simple fact that the poliovirus particles are so small (at 30 nm diameter they are hardly larger than a ribosome, and smaller than virus capsids previously averaged in situ by STA) and so smooth (no spikes or other low-res features to help drive alignment). We thus believe that the current tomograms don’t have enough signal to drive the alignment of isolated capsids to the point that we can start to see secondary structure.

- The presence of the membrane associated half-capsid intermediate is a very intriguing finding of this study. While I realize that there is a small number of subtomogram volumes trapping this assembly stage, I wonder if the authors have tried to see if there is any symmetry related information in these

particles. Is a 5-fold axis perpendicular to the membrane? Using the icosahedral reconstruction would be possible to obtain a missing wedge corrected cross correlation value for each of these volumes.
- The same question regarding the tethered capsids.

This is an important question and one that we tried to answer in response to this comment. We indeed believe that capsid intermediates and tethered capsids interact with membranes and tethers at specific sites. But we ran into a version of the problems outline above: the capsids don't seem to contain enough shape signal (i.e. too small and too smooth) to do this measurement. As an example of what we tried, below is the outcome when we aligned all individual tethered particles to a single-particle enterovirus structure. The red dots representing the tether position appear nearly randomly distributed which we attribute to insufficient signal.

Response Figure 2: Identified tether position (red dots) by cross-correlation of individual tethered particle subtomograms with a poliovirus reference (grey).

- Did the authors attempt the reconstruction of the tethering domain? While the resolution is very likely to be embarrassingly modest, it could offer some information regarding the dimensions of this feature.

We are not embarrassed by low resolution if the biological insights are good, and thus took the reviewer up on this excellent suggestion. We managed to pick and average ~200 tethered particles from the tomograms. Although the resolution of the average is indeed limited by particle number, the average provides an envelope that allows us to estimate the dimensions and molecular mass of the tether. We feel that this provides an important initial characterisation of the tether and added it to the new Fig. 1F, and lines 96-98.

- Is there any information behind the hypothesis that the granular material is actually RNA? Some EELS data of HPF-FS sections or the use of a dsRNA antibody could answer this question.

We tried to pursue this question by HPF-FS and immunolabelling. However, even with protocols optimised to retain the morphology of other autophagic structures (p62 assemblies) we could not distinguish different types of autophagosome contents (neither single virions, granules, nor the filament bundles). Thus, attempts at immunolabelling would make no sense. We have altered the text on lines 283-284 to reflect the uncertainty of the identity of the granules.